# The concurrence of DNA methylation and demethylation is associated with transcription regulation

Jiejun Shi [1], Jianfeng Xu[2], Yiling Elaine Chen [3], Jason Sheng Li [1], Ya Cui [1], Lanlan Shen [4], Jingyi Jessica Li [3] & Wei Li [1✉]

The mammalian DNA methylome is formed by two antagonizing processes, methylation by DNA methyltransferases (DNMT) and demethylation by ten-eleven translocation (TET) dioxygenases. Although the dynamics of either methylation or demethylation have been intensively studied in the past decade, the direct effects of their interaction on gene expression remain elusive. Here, we quantify the concurrence of DNA methylation and demethylation by the percentage of unmethylated CpGs within a partially methylated read from bisulfite sequencing. After verifying 'methylation concurrence' by its strong association with the co-localization of DNMT and TET enzymes, we observe that methylation concurrence is strongly correlated with gene expression. Notably, elevated methylation concurrence in tumors is associated with the repression of 40~60% of tumor suppressor genes, which cannot be explained by promoter hypermethylation alone. Furthermore, methylation concurrence can be used to stratify large undermethylated regions with negligible differences in average methylation into two subgroups with distinct chromatin accessibility and gene regulation patterns. Together, methylation concurrence represents a unique methylation metric important for transcription regulation and is distinct from conventional metrics, such as average methylation and methylation variation.

[1] Division of Computational Biomedicine, Department of Biological Chemistry, School of Medicine, University of California, Irvine, Irvine, CA, USA. [2] Department of Molecular and Cellular Biology, Baylor College of Medicine, Houston, TX, USA. [3] Department of Statistics, University of California, Los Angeles, CA, USA. [4] Department of Pediatrics, Baylor College of Medicine, USDA/ARS Children's Nutrition Research Center, Houston, TX, USA. ✉email: wei.li@uci.edu

DNA methylation at CpG dinucleotide(5mC) is introduced and maintained by DNA methyltransferases (DNMT family)[1,2]. Meanwhile, through hydroxymethylation, 5mC is removed by 10–11 translocation dioxygenases (TET family)[3,4]. Besides their opposing effects, the two enzyme families present complementary DNA-binding patterns. While TET1 protein prevents de novo methyltransferases from binding to regulatory elements[5,6], DNMT3A also blocks TET1 binding, especially in promoter regions[6]. Interestingly, these two 'competing' enzyme families are observed to be jointly associated with tumor malignancy. For example, in a conditional knockout study in the mouse hematopoietic system, the *Dnmt3a* and *Tet2* double-knockout mice show worse survival than single-knockout counterparts[7]; notably, mutations in *DNMT3A* and *TET2* also significantly co-occur in human T-cell lymphoma[8]. These findings suggest that the concurrence of methylation and demethylation processes is related to tumorigenesis. However, to what extent this concurrence contributes to cancer gene regulation remains largely unknown.

For years, DNA methylation levels have been quantified in an 'average' manner. The increased average methylation level of CpG islands (CGIs), i.e., CGI hypermethylation, is a well-established mechanism for gene silencing[9]. Numerous differentially methylated regions have been identified based on the between-sample comparison of average methylation levels[10,11]. Besides the average methylation, DNA methylation has been quantified by its variation as 'methylation heterogeneity'[12] or 'epigenetic polymorphism'[13]. Methylation heterogeneity scores are defined based on the frequencies of methylation patterns (epialleles) at multiple CpGs inferred from bisulfite sequencing reads[12–15]. Previous studies reveal that methylation variation is associated with global transcription variation[12,16,17]. However, to capture more accurate heterogeneity information, these methylation variation quantifications require at least 4 CpGs covered by each sequencing read, and thus can only utilize ~20% of total reads in the genome (Supplementary Fig. 1). Moreover, neither average methylation nor methylation variation can delineate the degree of concurrence between active methylation and demethylation.

Recently, a mathematical model has been applied to deconvolute methylation and demethylation rates from average methylation levels of individual CpGs in stem cells[18]. However, the spatial coupling of methylation concurrence at adjacent CpGs has not been considered, and such coupling may be critical for transcription factor (TF) binding and cancer gene regulation[19].

Bisulfite sequencing has enabled the measurement of DNA methylation of adjacent CpGs within the same read[20], and thus is able to capture methylation concurrence if there exist unmethylated CpGs in a partially methylated read. We demonstrate that methylation concurrence unveils a unique type of methylation abnormality, which is distinct from both the change of average methylation and methylation variation in many aspects. We find that methylation concurrence is associated with a previously undetected repertoire of epigenetically regulated tumor suppressor genes (TSGs), and that it can be used to stratify large undermethylated regions into two subgroups with distinct characteristics in chromatin accessibility and gene regulation.

## Results

### Delineating the concurrence of active DNA methylation and demethylation.
We quantify the concurrence of active DNA methylation and demethylation within the same cell by dissecting reads from bisulfite sequencing. The methylation concurrence events are captured by the unmethylated CpG(s) within a partially methylated read (red circles in Fig. 1a) because each read comes from one cell. Fully methylated and unmethylated reads, in contrast, do not possess information on DNMT and TET concurrence, as they are dominated by methylation and demethylation, respectively. Hence, we dissect bisulfite sequencing reads into three categories of fragments (or sub-reads), i.e., methylated fragments which consist of consecutive methylated CpG(s) (solid circles in Fig. 1a, denoted as 'M'), unmethylated fragments which are the fully unmethylated reads (blank circles, denoted as 'U'), and methylation-concurrence fragments which are segments of unmethylated CpG(s) in partially methylated reads (red circles, denoted as 'C'). We define the 'methylation concurrence ratio' of a genomic region as the sum of methylation-concurrence fragments' weights divided by the sum of all fragments' weights in that region (Eq. (1) in Methods). Each fragment's weight is set as its number of CpGs (see Methods).

We compare the methylation concurrence ratio with two measures of the average methylation (i.e., the traditional mean methylation and the cellular heterogeneity-adjusted clonal methylation (CHALM)[21], see Methods) and three measures of the methylation variation (i.e., Shannon's entropy[15], Epipolymorphism[13], and the proportion of discordant reads (PDR)[12], see Methods). Supplementary Figure 1b compares the calculations of different metrics with simulated data. The methylation variation scores are window-based and only take reads covering at least 4 CpGs (only ~20% of total reads), while methylation concurrence has no such limitation and utilizes all reads (Supplementary Fig. 1c). Furthermore, methylation concurrence detects more regulatory elements (e.g., promoters) than methylation variation (Supplementary Fig. 1d).

Using whole-genome bisulfite sequencing (WGBS) data from mouse embryonic stem cells (mESCs), we observe that these six methylation measures are correlated to different degrees with DNA-binding intensities of DNMT3A1 and TET1 enzymes at gene promoters in matched samples measured by ChIP-seq (chromatin immunoprecipitation followed by high-throughput sequencing). We observe that the average methylation measures are correlated positively and negatively with the binding intensities of the methyltransferase DNMT3A1 and the demethylase TET1, respectively, consistent with the known enzymatic activities of the two enzymes (Supplementary Fig. 2b and 2c). The three methylation variation measures show similar correlation patterns (Supplementary Fig. 2d, 2e, and 2f). In contrast, the methylation concurrence ratio is positively correlated with both DNMT3A1 and TET1 binding intensities (Supplementary Fig. 2a). To further explore the two enzymes' joint effects, we define the DNMT3A1-TET1 'joint regulation score' (Π) of a promoter as the product of DNMT3A1 and TET1 binding intensities within that promoter (see Eq. (2) in Methods). This joint regulation score depicts the extent to which the promoter is co-occupied by both enzymes, and it takes a low value if either enzyme has a low binding intensity. As expected, the joint regulation score has a strong positive correlation with the methylation concurrence ratio (Fig. 1b) but not as much with the average methylation measures (Fig. 1c and Supplementary Fig. 2c) or the methylation variation measures (Fig. 1d and Supplementary Fig. 2e and 2f). This result is consistent with the fact that average methylation and methylation variation are neutralized by the additive effects of two opposing enzymes and that only the methylation concurrence ratio characterizes the concurrence of DNMT3A1 and TET1. In the following text, we use the traditional mean methylation as the average methylation measure and Shannon's entropy as the methylation variation measure.

Based on DNMT3A1 and TET1 colocalization patterns, we categorize promoters into four groups: DNMT3A + TET1+, DNMT3A + TET1−, DNMT3A − TET1+, and DNMT3A − TET1−, where + and − indicate strong and weak binding, respectively. As shown in Fig. 1e, the methylation concurrence

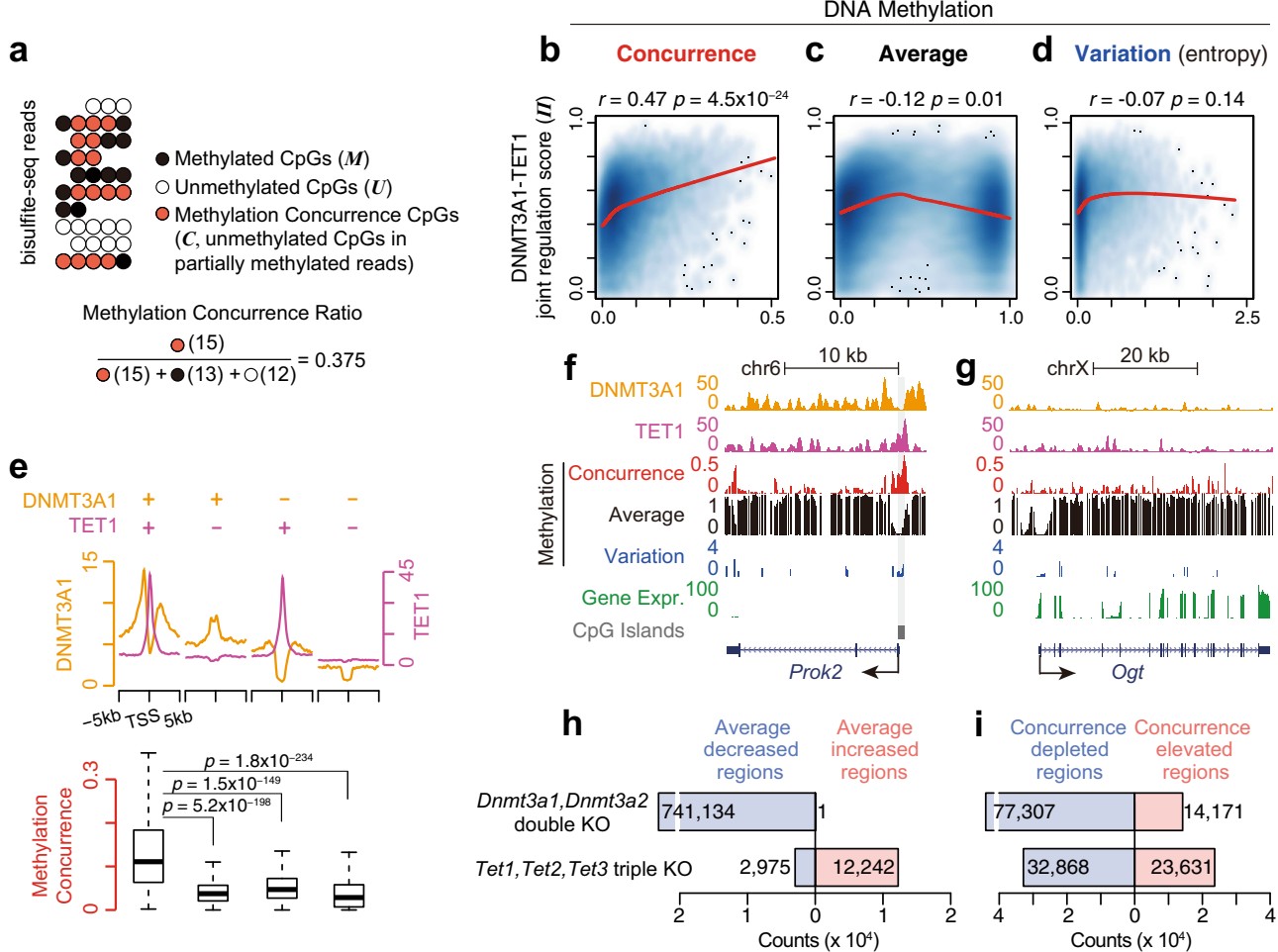

**Fig. 1 The methylation concurrence ratio measures the antagonism between methylation and demethylation processes in mESCs. a** Schematic of DNA methylation concurrence captured by bisulfite-seq. Solid circles are methylated cytosines. Blank circles are unmethylated cytosines. Red circles are unmethylated cytosines in partially methylated reads, i.e., methylation- concurrence cytosines. The equation below shows the calculation of methylation concurrence ratio using the example above. **b** Methylation concurrence is positively correlated with the 'DNMT3A1-TET1 joint regulation score' ($\Pi$) in gene promoter regions. Average methylation (**c**) and methylation variation (**d**) are not correlated with $\Pi$ at gene promoters. Spearman's rank correlation was calculated. $P$ values were calculated by the two-tailed correlation test for Spearman's correlation. LOWESS lines were plotted to describe the relationships between variables (indicated by red curves). **e** The methylation concurrence ratio is significantly higher at DNMT3A1&TET1 co-occupied promoters. Gene numbers of each groups: 'DNMT3A1 + TET1+', $n = 1294$; 'DNMT3A1 + TET1 − ', $n = 1238$; 'DNMT3A1−;TET1+', $n = 1370$; 'DNMT3A1 − TET1 − ', $n = 1595$. The two-tailed Mann–Whitney U test was used for the significance test. The line in the box center refers to the median, the limits of box refer to the 25th and 75th percentiles and whiskers are plotted at the highest and lowest points within the 1.5 times interquartile range. **f** UCSC Genome Browser tracks show DNMT3A1 binding (orange), TET1 binding (purple), methylation concurrence (red), average methylation (black), methylation variation (blue), and gene expression data (green) at *Prok2* gene. CpG islands are shown in gray. **g** Same as (**f**), but for gene *Ogt*. **h** *Dnmt3a* knockout leads to a decrease in average methylation, while *Tet* knockout leads to hypermethylation. **i** Both *Dnmt3a* knockout and *Tet* knockout lead to more concurrence depletion than elevation. According to the original paper, the '*Dnmt3a1, Dnmt3a2* double knockout' sample is generated by reintroducing DNMT3B1 into stem cells that lack DNA methylation due to deletions of all *Dnmt* genes (*Dnmt3a1, Dnmt3a2*, and *Dnmt3b1*). The genomic binding of the reintroduced DNMT3B1 in knockout cells resembles that in wild-type ES cells[22].

ratio is significantly higher in 'DNMT3A + TET1 + ' promoters than in the other three groups of promoters, which are bound by only one or none of the two enzymes. For example, *Prok2* (Fig. 1f and Supplementary Fig. 3a) has a DNMT3A + TET1+ promoter and a high methylation concurrence ratio, while *Ogt* (Fig. 1g and Supplementary Fig. 3b) has a DNMT3A − TET1− promoter and a low methylation concurrence ratio. Notably, only the methylation concurrence ratio can distinguish the different colocalization patterns of *Prok2* and *Ogt*, while the average methylation and the methylation variation cannot. Moreover, only the methylation concurrence ratio is predictive of the expression levels of *Prok2* and *Ogt* (*Prok2* has a high ratio and low expression, while *Ogt* has a low ratio and high expression), while the average methylation

and the methylation variation are not (both genes have similar average methylation and methylation variation levels).

In addition to DNMT3A and TET1, the methylation concurrence ratio correlates with other methylation/demethylation enzymes, such as DNMT3B (Supplementary Fig. 4a) and TET2 (Supplementary Fig. 4b). Furthermore, the methylation concurrence ratios are also positively correlated with the joint regulation scores of additional combinations between the methylation/demethylation enzymes, such as DNMT3A1-TET2, DNMT3B-TET1, and DNMT3B-TET2 (Supplementary Fig. 4c, d, e).

These findings are supported by additional evidence in mouse samples with *Dnmt* or *Tet* knockout. As expected, *Dnmt3a1* and *Dnmt3a2* double-knockout[22] leads to genome-wide hypomethylation

in mouse ESC, whereas *Tet1, Tet2, Tet3* triple-knockout leads to global hypermethylation (Fig. 1h). We observe that either knockout results in more regions with decreased methylation concurrence ratios than regions with increased ratios, an observation consistent with our definition of methylation concurrence ratio (Fig. 1i). The knockout experiments in human ESC[23] also reach a similar conclusion. As in Supplementary Fig. 5a, *DNMT3A* and *DNMT3B* double-knockout (DKO) leads to a substantial decrease of average methylation, while *TET1, TET2, TET3* triple-knockout (TKO) leads to hypermethylation. The *DNMT3A, DNMT3B, TET1, TET2,* and *TET3* pentuple knockout (PKO) sample shows a relatively 'balanced' change of average methylation. In line with our previous findings in mouse ESC (Fig. 1i), there are more depleted regions than elevated regions of methylation concurrence in DKO (Supplementary Fig. 5b). In agreement with the mouse data, methylation concurrence appears more sensitive to changes in DNMT activity than TET activity, suggesting additional mechanisms may be involved in its negative regulation. For example, some unmethylated regions in WT become partially methylated in TKO, so the methylation concurrence is elevated rather than depleted (Supplementary Fig. 5c). Intriguingly, as indicated in the bottom row of Supplementary Fig. 5b, the knockout of both enzyme families (PKO) leads to a dominant trend of depleted concurrence. Overall, these results confirm that the methylation concurrence ratio, an emerging quantitative measure of methylation, can delineate the antagonism between methylation and demethylation processes.

**Methylation concurrence is negatively correlated with gene expression**. Our previous analysis of genes *Prok2* and *Ogt* (Fig. 1f and 1g) suggests that the methylation concurrence ratio may be a better predictor of gene expression than the average methylation and the methylation variation. To further examine the relationship between these quantitative measures of methylation and gene expression, we first calculate the methylation concurrence ratios, the average methylation, and the methylation variation of every CpG site as well as transcription regulatory elements/regions of three types (promoters, gene-body regions, and enhancers, see Methods) using WGBS data of primary cells and normal tissues from Epigenomic Roadmap Consortium[24]. Then, we quantify gene expression levels using the RNA-sequencing (RNA-seq) data from matched samples (Supplementary Data 1) and separate all genes into four equal-sized groups based on expression quantiles (0–25%, 25–50%, 50–75%, and 75–100%). As indicated in Fig. 2a and Supplementary Fig. 6a, more highly expressed genes are characterized by lower methylation concurrence ratios, lower average methylation (by both the traditional mean and CHALM), and lower methylation variation in promoters. However, unlike the average methylation and the methylation variation, the methylation concurrence ratios in TSS-proximal regions (<2 kb) are higher than in more distal regions (>2 kb), suggesting that the antagonism between methylation and demethylation is more intense near transcription start sites. Our findings are in line with a previous knockout study, which reveals that DNMT3A and TET1 prevent binding of each other mainly in TSS-proximal regions[6].

Compared with the average methylation (measured by both the traditional mean and CHALM) and the methylation variation, the methylation concurrence ratio is better correlated with gene expression, and this phenomenon is consistent across all three types of regulatory elements in CD3 + T cells (Fig. 2b and Supplementary Fig. 6b). In contrast, the correlations between gene expression and the average methylation or the methylation variation are not consistent across the three types of regulatory elements; for example, CHALM values in gene-body regions and enhancers have much lower correlations with gene expression

than CHALM values in promoters do (Supplementary Fig. 6b). We further confirm this phenomenon using WGBS and RNA-seq data of other primary cells, fetal tissues, and adult tissues (Supplementary Fig. 7).

In addition to being associated with baseline gene expression level, the differences (Δ) of methylation concurrence are also negatively correlated with gene expression changes (Supplementary Fig. 8a) across its full dynamic range. In contrast, the differences of average methylation (Supplementary Fig. 8b) and methylation variation (Supplementary Fig. 8c) only show a negative correlation in half of their dynamic range (Δ > 0). Altogether, the methylation concurrence ratio is a more unbiased predictor of gene expression than both the average methylation and the methylation variation.

**Effects of sequencing depth and CpG density on quantification of methylation concurrence**. Sequencing depth is a key factor that affects quantitative analysis of high-throughput sequencing data. Using a down-sampling strategy, we observe that all of the three measures (methylation concurrence, average methylation, and methylation variation) are negatively associated with gene expression at sufficient sequencing depth (Supplementary Fig. 9a). Notably, across sequencing depths from ~4× to ~86×, the methylation concurrence ratio has consistently better correlations with gene expression than the average methylation and the methylation variation.

Besides sequencing depth, CpG density also affects the correlations between methylation measures and gene expression. To investigate this issue, we stratify genes into three groups based on the CpG densities in their promoters, i.e., high-CpG promoter (HCP) genes, intermediate-CpG promoter (ICP) genes, and low-CpG promoter (LCP) genes[25]. For all three methylation measures (the methylation concurrence ratio, the average methylation, and the methylation variation), the correlations between their values in promoters and gene expression decrease from HCP genes to ICP genes and further down to LCP genes. Among the three measures, the methylation concurrence ratio consistently has the strongest negative correlation with gene expression for all three groups of genes (Supplementary Fig. 9b).

**Methylation concurrence is associated with the repression of tumor suppressor genes**. To investigate whether methylation concurrence is involved in tumorigenesis, we apply gene set enrichment analysis (GSEA) to profiles of promoter methylation concurrence ratios in 8 TCGA normal samples. As shown in Supplementary Fig. 10a and 10b, the curated tumor suppressor genes (TSGs) in the COSMIC Cancer Gene Census (CGC) database[26] are associated with low methylation concurrence ratios. In contrast, cell lineage genes such as Homeobox genes are associated with high methylation concurrence ratios. In addition, housekeeping functions such as 'pentose phosphate pathway' do not exhibit associations with methylation concurrence. Further analysis of 56 normal samples' methylomes shows a conserved pattern of low methylation concurrence at the promoter of *TP53*, a well-known TSG (Supplementary Fig. 10c), and high methylation concurrence at the promoter of *NKX1-1*, a Homeobox transcription factor related to organ development and regeneration[27] (Supplementary Fig. 10d).

Promoter hypermethylation is a well-established mechanism for TSG silencing in tumors[28]. We compare the three methylation measures between the normal and tumor uterus samples from the same patient in TCGA. Using local-FDR, a statistical criterion that assesses the credibility of individual discoveries under the Bayesian framework[29], we identify 2557 genes, 2056 genes, and 871 genes whose promoters have significant changes of average

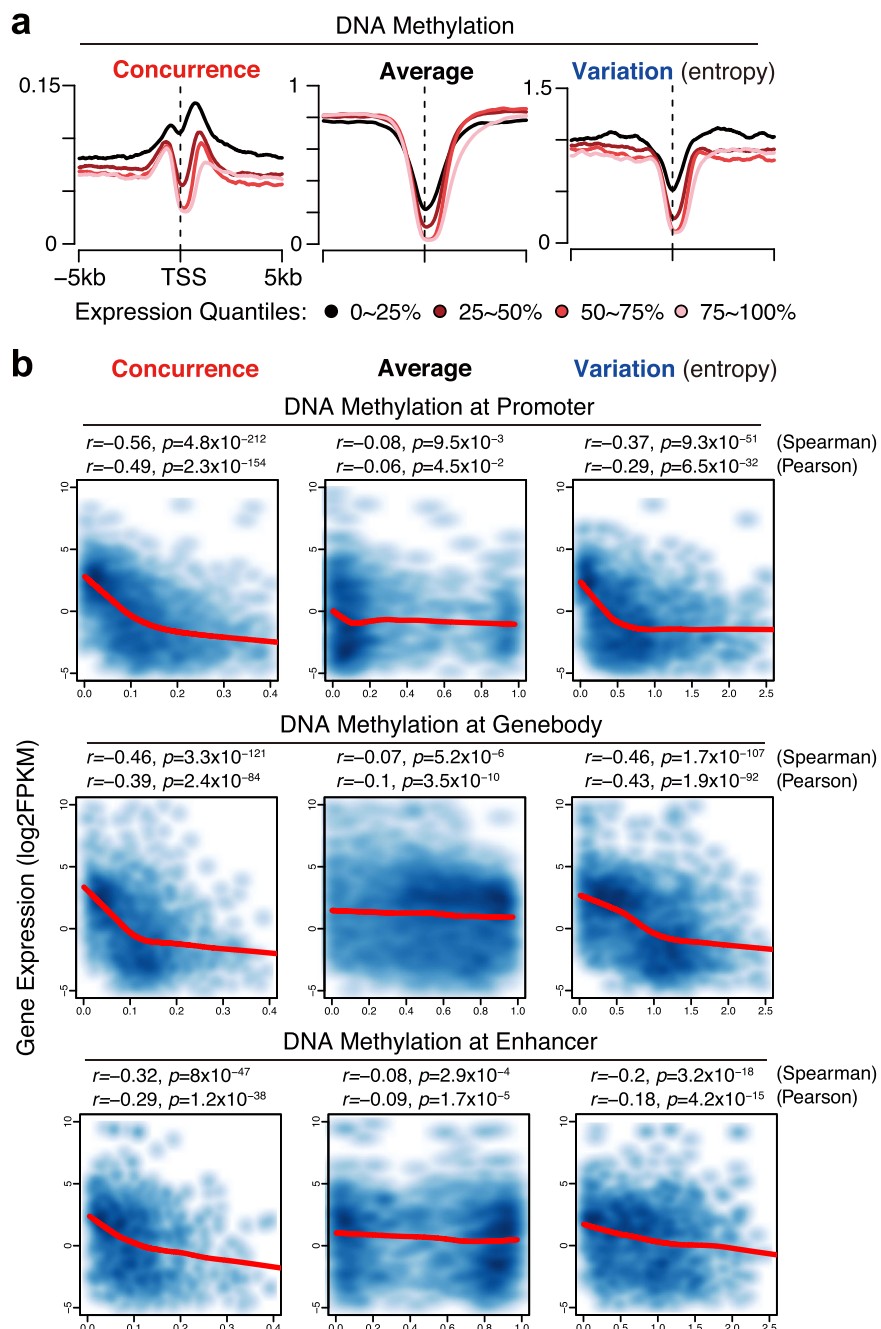

**Fig. 2 Methylation concurrence is negatively associated with gene expression. a** Average profiles in TSS-proximal regions of four gene groups, divided by quantiles of expression levels, in CD3+ T cells. Methylation-concurrence ratios are on the left, average methylation ratios are in the middle, and methylation variation scores(entropy) are shown on the right. 0–25%, lowest expressed; 25–50%, lower expressed; 50–75%, higher expressed; 75–100%, highest expressed. The gene number of each group is 3600. **b** The promoter/gene-body/enhancer methylation concurrence ratios (1st column) are strongly negatively correlated with gene expression level in CD3+ T cells, and this correlation is stronger than that of the average methylation (2nd column) and the methylation variation (3rd column). Promoter regions are from 1 kb upstream to 500 bp downstream of TSS. Gene-body regions are from 500 bp downstream of TSS to TTS. Enhancer regions are defined based on chromatin interactions validated by Hi-C data (see Methods). The WGBS data are from the Roadmap project with GEO accession number GSM1186660. The CD3+ primary cells are from a 37-year-old male. To increase reliability, we select regulatory elements whose CpGs are all sufficiently covered (≥4 reads). To make fair comparisons, only the elements which can be detected by all three metrics are included. This results in 9876 promoter regions, 14,868 gene-body regions, and 3351 enhancer regions. Spearman's rank correlation and Pearson's correlation were calculated based on all data points. *P* values were calculated by the two-tailed correlation test. LOWESS lines were plotted to describe the relationships between variables (indicated by red curves).

methylation, methylation concurrence, and methylation variation. Through an overlap analysis between methylation concurrence and average methylation, we identify three gene groups: 487 methylation-concurrence elevated and average-methylation stable genes (P1), 842 methylation-concurrence stable and average methylation increased (hypermethylated) genes (P2), and 265 methylation-concurrence elevated and average methylation increased (hypermethylated) genes (P3) (Fig. 3a). For the three

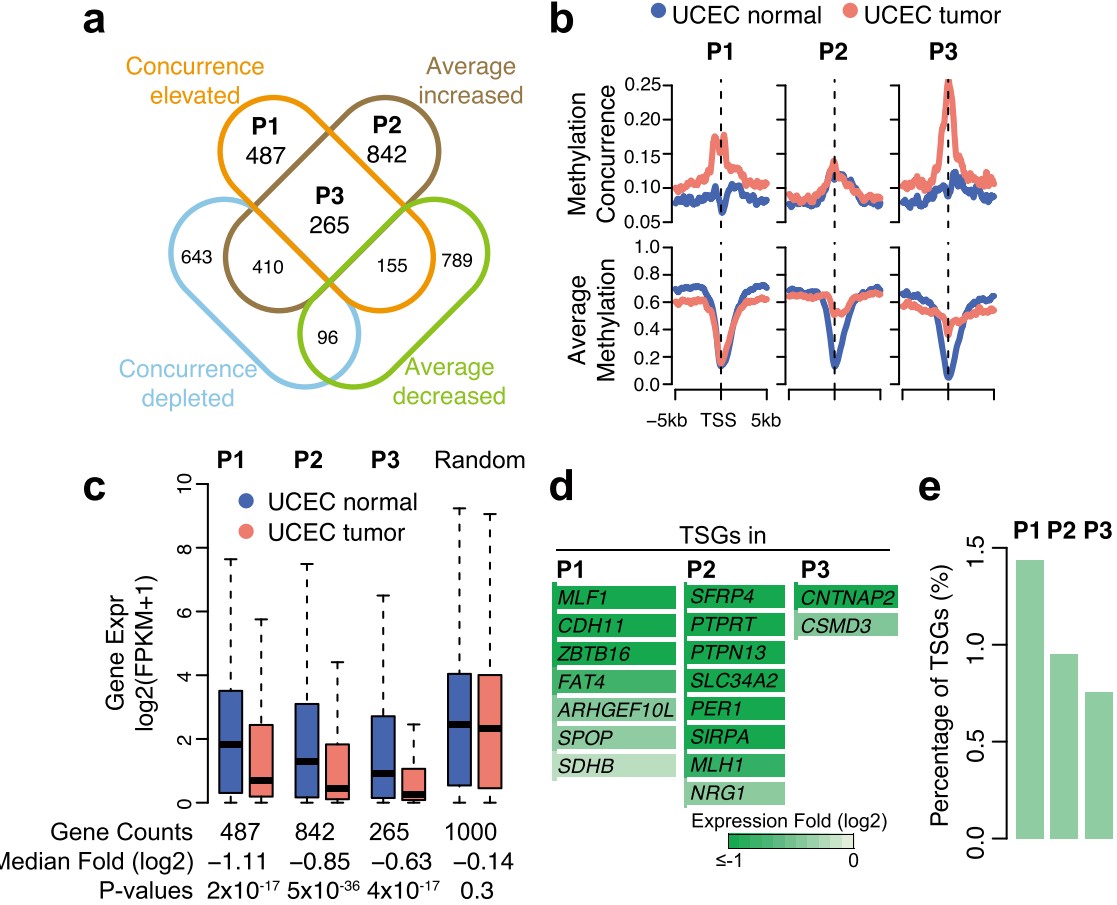

**Fig. 3 Tumor repressors can be repressed by methylation concurrence elevation rather than hypermethylation in uterus tumor. a** Venn diagram shows the overlap between promoter methylation concurrence altered genes and average methylation altered genes in TCGA uterus tumor (UCEC). P1, methylation concurrence elevated but average methylation stable genes; P2, hypermethylated but concurrence-stable genes; P3, concurrence-elevated and hypermethylated genes. **b** Average profiles of methylation concurrence (upper panel) and average methylation (lower panel) in TSS regions of P1, P2, and P3 genes. UCEC normal is in blue, and UCEC tumor is in red. **c** Gene expression change of P1, P2, P3, and 1000 randomly selected genes. The fold changes between median values (log2 scale) are indicated below. The two-tailed Wilcoxon signed-rank test was used for the significance test. The line in the box center refers to the median, the limits of box refer to the 25th and 75th percentiles and whiskers are plotted at the highest and lowest points within the 1.5 times interquartile range. The lists of repressed genes in three groups are in Supplementary Data 2. **d** COSMIC tumor suppressor genes which are overlapped with P1, P2, and P3 genes. Gene expression fold changes (log2 scale) are indicated. Darker green signifies higher repression in the UCEC tumor sample. **e** The percentages of tumor suppressors in P1, P2, and P3 genes.

gene groups, the average profiles of the methylation concurrence ratios and the average methylation in TSS-proximal regions are illustrated in Fig. 3b. Compared with randomly selected genes, all three groups of genes demonstrate more significant transcriptional repression (i.e., smaller *p*-values) in the tumor uterus sample compared with the matched normal uterus sample by the Wilcoxon signed-rank test (Fig. 3c). We find 17 TSGs in the three gene groups, and 41% (7/17) of them cannot be explained by promoter hypermethylation alone (P1 TSGs in Fig. 3d). For example, *ZBTB16* (also known as PLZF) is a P1 TSG that inhibits prostate cancer tumor growth through its interplay with *PTEN* and *FOXO3a*[30], and its genetic alterations have been found in metastatic prostate cancer samples[31]. Although *ZBTB16* has an unclear function for uterine cancer, a previous study revealed that overexpression of *ZBTB16* inhibited proliferation in cervical carcinoma cells and induced apoptosis[32]. Given the visible association between high methylation concurrence in promoter regions and reduced gene expression (Supplementary Fig. 11a), these results suggest that *ZBTB16* may also act as a tumor suppressor in uterine tumors, although additional functional studies are needed to test this hypothesis.

To check whether methylation variation (i.e., the methylation entropy) is able to explain TSG repression, we perform a similar overlap analysis between methylation entropy and average methylation. We find that the 214 entropy elevated and average-methylation stable genes (Q1, Supplementary Fig. 11b) are not significantly repressed in the tumor uterus sample compared to the matched normal sample (Supplementary Fig. 11d, *p* value = 0.2), and that the single known TSG (*MLF1*, Supplementary Fig. 11e) in the Q1 group can also be detected by methylation concurrence (Fig. 3d). This result shows that methylation concurrence better associates with the repression of TSGs compared to methylation entropy.

We also analyze the methylomes of normal and tumor breast samples from the same patient in TCGA (Supplementary Fig. 12). The overlap analysis indicates that 63% (7/11) of repressed TSGs can be explained by elevated methylation concurrence but not by hypermethylation, and *ZBTB16* is again an example (Supplementary Fig. 12d). Collectively, our analysis suggests that the methylation concurrence reveals a repertoire of epigenetically regulated tumor suppressor genes that cannot be detected by the average methylation or the methylation variation.

**Methylation concurrence in large undermethylated regions**. Undermethylated regions (UMRs) are the clusters of adjacent, lowly methylated CpGs spanning from hundreds to thousands of base pairs[33]. Large UMRs, which are also termed as methylation 'canyons'[33] or 'valleys'[34], are conserved across cell types[33] and involved in chromatin interactions[35], and their target genes are associated with cell lineages[34]. Although all of the methylation canyons are poorly methylated (with the average methylation ≤0.1), their methylation concurrence ratios range from 0 to 0.4 (Supplementary Fig. 13d). To examine the role of methylation concurrence in their function, we classify the methylation canyons into high-concurrence canyons (blue in Fig. 4) and low-concurrence canyons (red in Fig. 4), based on a cutoff derived from the genome background (see Supplementary Fig. 13 and Methods). Although the two groups do not differ in their average methylation, their chromatin accessibilities are dramatically distinct (Fig. 4a). Accordingly, we refer to the low-concurrence canyons, which are enriched with active markers (H3K4me3, H3K27ac, H3K36me3, and DNase I hypersensitive sites), as active canyons (aCanyons); we refer to the high-concurrence canyons, which are enriched with H3K27me3, as Polycomb canyons (pCanyons). To validate their distinct chromatin activities, we perform the same analysis in human embryonic stem cell H1, for which ChIP-seq data of 28 types of histone modifications/variants are available. Similar to the case in CD3 + T cells, active markers (H3K4me3, H3K27ac, H2A.Z, H3K9ac, etc.) in H1 ESCs are more enriched in aCanyons. In contrast, the repressive markers (H3K27me3, H3K9me3, etc.) in H1 are more enriched in pCanyons (Supplementary Fig. 14).

Although the methylation variation measures (entropy, Epipolymorphism, and PDR) also differ between aCanyons and pCanyons (Supplementary Fig. 15a), they do not provide the same information as the methylation concurrence ratio does: only the methylation concurrence ratio is higher inside pCanyon than in the flanking regions. This phenomenon is consistent with the finding of a previous study that TET1 binding is elevated in canyons in *Dnmt3a* knockout sample[6]. In addition, the average differences (Δ) of methylation concurrence between pCanyons and aCanyons are more substantial than for methylation variation scores. The Δmethylation-concurrence inside Canyon is ~2.8-fold higher than the Δmethylation-concurrence in Canyon flanking regions. For methylation variation, this fold decreases to ~1.5 for ΔEntropy, ~1.8 for ΔEpipolymorphism, and ~1.3 for ΔPDR (Supplementary Fig. 15b). These data indicate methylation concurrence can better distinguish aCanyons from pCanyons.

Furthermore, the TF binding patterns are consistent with the definitions of aCanyons and pCanyons. The aCanyons are bound by TFs involved in active transcription, such as transcription initiation factor TFIID subunit 1 (TAF1), TATA-box binding protein (TBP), and RBBP5, a subunit of MLL complex. In contrast, the pCanyons are bound by subunits of the PRC2 complex, such as SUZ12 and EZH2 (Supplementary Fig. 16a). To exclude bias that may be introduced by Canyon length differences (Supplementary Fig. 16b), we perform the same analysis for the Canyon gene promoters, which feature the same length for different genes. The results confirm the different TF binding preferences in aCanyon gene promoters and pCanyon gene promoters (Supplementary Fig. 16c). Recently, Zhang et al. revealed that methylation canyons are involved in chromatin loops that rely on Polycomb binding instead of cohesion or CTCF[35]. Consistent with this finding, we find that pCanyons are more enriched with chromatin interactions than aCanyons (Supplementary Fig. 16d).

To gain further insights into the functions of canyons, we define canyon targets as the genes whose promoters or gene-body regions overlap with canyons. Again, consistent with the definitions of aCanyons and pCanyons, aCanyon targets are highly expressed, while pCanyon targets are almost silenced (Fig. 4b). Gene ontology analysis reveals that aCanyon targets are enriched with TSGs and cancer pathways, while pCanyon targets are enriched with cell fate commitment and Homeobox genes. Both negative control (Ctrl) groups of non-Canyon-target genes —Ctrl group with similar expression distributions as aCanyon targets and Ctrl group of randomly selected lowly expressed genes —are not enriched with these functional terms (Fig. 4c).

Finally, we correlate the expression of canyon targets with the average methylation in a locus-specific manner[36]. Specifically, we first divide the promoter and downstream region (−2 to +10 kb) of each gene into 120 equal-length bins. Then we compute the correlation between gene expression and the average methylation in each bin across aCanyon targets, pCanyon targets, and Ctrl genes. As expected, the average methylation in promoters is negatively correlated with gene expression for all four gene sets, albeit to varying degrees (Fig. 4d). Previous studies also report positive correlations between the average methylation in gene-body regions and gene expression[10,37]. However, we observe that this phenomenon occurs for pCanyon targets, but not for aCaynon targets (Fig. 4d). Furthermore, although both highly expressed (Fig. 4b) and highly methylated at gene-body (Fig. 4e), the aCanyon targets present a unique regulating model compared to the higher expressed Ctrl group (Fig. 4d). The surprisingly negative correlations between the average methylation in gene-body regions and gene expression for aCanyon targets suggest that these genes may have a distinct methylation regulation mechanism.

Using in vitro SELEX assays, Yin et al. profile human TFs for binding preferences towards 5mC[38]. They classify these TFs as 'methyl-plus' and 'methyl-minus' groups, such that methyl-plus TFs' binding is enhanced by 5mC, while methyl-minus TFs' binding is inhibited by 5mC. To further understand aCanyons and pCanyons, we utilize this SELEX data and perform differential analysis of TF motif enrichment. We find that the TF motifs' 5mC preferences are negatively correlated with their fold enrichment between aCanyon and pCanyon (Fig. 4g). In other words, methyl-plus TF motifs are more enriched in pCanyons, while methyl-minus TF motifs are more enriched in aCanyons. Therefore, when methylation increases, the aCanyons would lose TF binding, and the aCanyon targets would have decreased expression. On the other hand, when methylation increases, the pCanyons would have more TF binding, and the pCanyon targets would have increased expression. To avoid the bias due to Canyon length differences (Supplementary Fig. 16b), we also perform the same analysis for the Canyon gene promoters. The results agree with previous observations in Canyons (Supplementary Fig. 16e). This model helps explain the distinct regulations of aCanyon and pCanyon targets in one particular cell type. However, how methylation concurrence captures their transcriptional changes from one condition to another is unclear and will need to be elucidated in follow-up investigations.

Together, methylation concurrence shows that the dynamics of methylation and demethylation are unique for aCanyons and pCanyons and strongly associate with additional distinguishing features (e.g., chromatin accessibility) of their regulation.

## Discussion

Cytosine methylation is a reversible biochemical modification[39]. The global pattern of mammalian methylome is formed by two antagonizing processes: methylation and demethylation. The methylation concurrence ratio defined in this paper utilizes

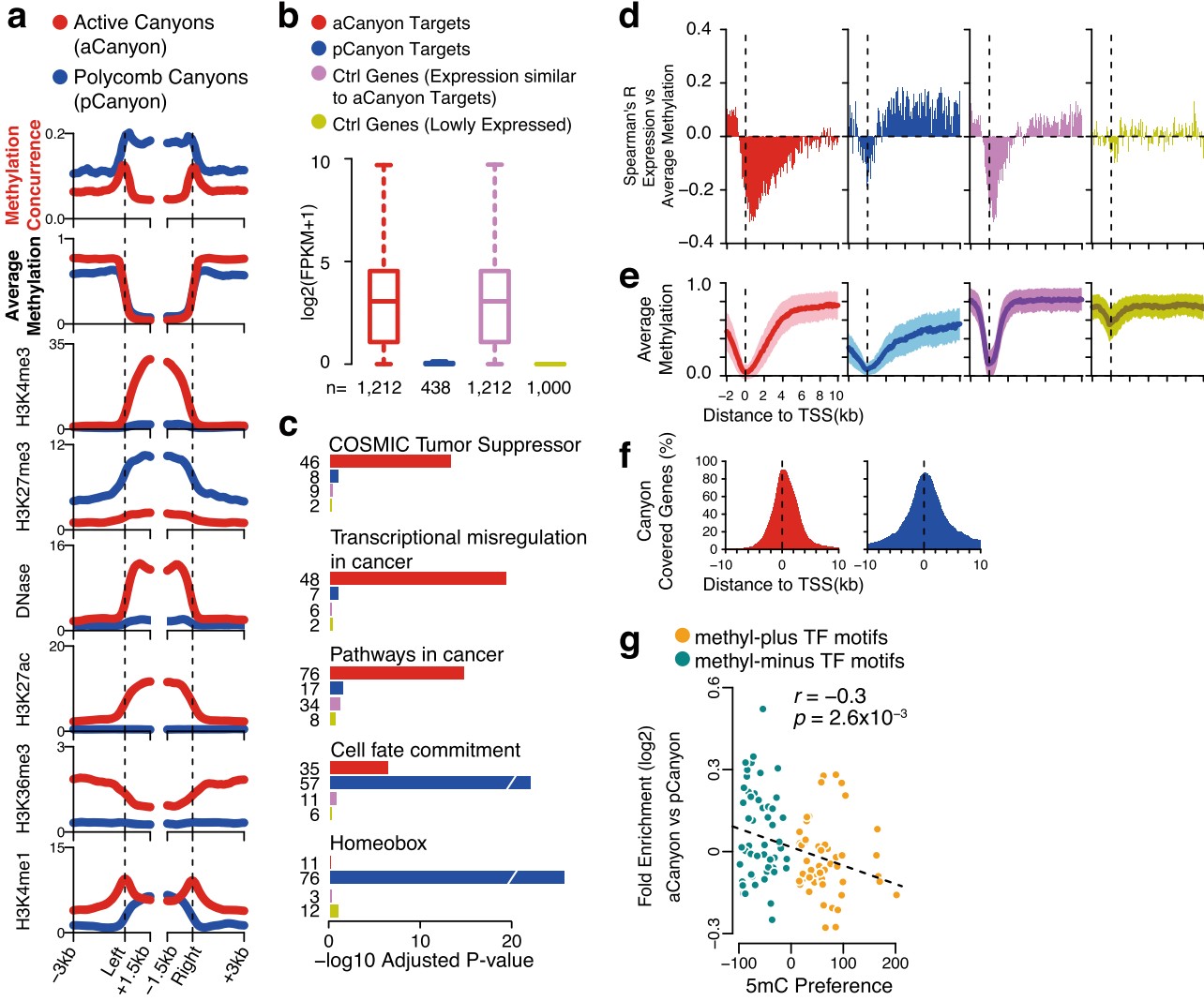

**Fig. 4 Distinct characteristics of methylation canyons categorized by methylation concurrence.** Average profiles of methylation concurrence, average methylation, DNase I hypersensitive sites, H3K4me3, and H3K27me3 in methylation canyons in CD3+ T cell. 'aCanyons' (red) are low-concurrence canyons. 'pCanyons' (blue) are high-concurrence canyons. The X-axis indicates the distance to canyon borders. **b** Expression of aCanyon target genes (red, n = 1212), pCanyon target genes (blue, n = 438), control group which features a similar expression distribution as aCanyon Targets (purple, n = 1212), and control group of randomly selected lowly expressed genes (gold, n = 1,000). The line in the box center refers to the median, the limits of box refer to the 25th and 75th percentiles and whiskers are plotted at the highest and lowest points within the 1.5 times interquartile range. Lists of aCanyons and pCanyons are in Supplementary Data 3, together with their target genes. **c** Functional enrichment analysis of aCanyon target genes, pCanyon target genes, and control groups. Enriched gene counts in each group are indicated on the left side. P-values were measured by two-tailed Fisher's Exact Test and adjusted by the Benjamini–Hochberg method. **d** Spearman correlation between gene expression and average methylation of 100 bp-bin in gene regions. In total, 120 bins from −2 kb to 10 kb were measured. **e** The profiles of average methylation for each gene group. The standard deviations are indicated by the width of the shaded area. **f** The Canyon distribution around the TSS. In each position from TSS−10 kb to TSS+10 kb, the percentage of aCanyon (red) and pCanyon (blue) target genes covered by Canyon is shown on the Y-axis. **g** Relationship between fold enrichment and 5mC preference of TF motifs. Each dot represents a motif. Y-axis indicates the fold change (log2) between enrichment at aCanyon and enrichment at pCanyon of the same motif (see Methods). The X-axis shows the 5mC preference of motifs measured by the SELEX technique. 'methyl-plus' TFs prefer to bind methylated sequences, while binding of 'methyl-minus' TFs are not favored by 5mC. The list of 'methyl-plus' and 'methyl-minus' TFs are in Supplementary Data 4. Spearman's rank correlation was used. P values were calculated by the two-tailed correlation test for Spearman's correlation. A linear model was plotted to describe the relationships between variables (indicated by the dashed line).

unmethylated fragments inside the partially methylated reads to measure this antagonism. Through reanalyzing bisulfite sequencing data, we reveal that the methylation concurrence ratio is strongly correlated with gene expression. Compared to the average methylation, the methylation concurrence ratio in different types of regulatory elements, e.g., promoter, gene-body, and enhancer, consistently has stronger correlations with gene expression. Another advantage of methylation concurrence is that it utilizes all reads, while the methylation variation scores are

window-based and only take reads covering at least 4 CpGs. To make a fairer comparison, we recalculate methylation concurrence using the same reads used by methylation variation. The results confirm that methylation concurrence still performs better than average methylation (Supplementary Fig. 19) and methylation variation (Fig. 1d and Fig. 2b). It is worth noting that although its global correlation with expression is superior to the other compared metrics, methylation concurrence is not a perfect indicator for all genes. Other metrics have unique value in

depicting methylation dynamics (Supplementary Fig. 20c) and sometimes show a similar trend with methylation concurrence (Fig. 2b and Supplementary Fig. 15). In the future, a better indicator may be proposed by integrating different methylation metrics. Overall, our analysis confirms the negative impact of methylation concurrence on gene transcription, which extends the interpretation of DNA methylation data.

In terms of cancer gene regulation, we find that TSGs are characterized by conserved low methylation concurrence ratios across normal samples. During tumorigenesis, the elevation of methylation concurrence serves as an emerging mechanism that can explain the repression of 40–60% of TSGs, which cannot be explained by hypermethylation. Thus, our definition of methylation concurrence expands our understanding of aberrant tumor methylation in addition to promoter hypermethylation.

There are limitations to the methylation concurrence metric. First, 5mC can be oxidized to 5-Hydroxymethylcytosine (5hmC) by the TET proteins, but the conventional bisulfite-seq cannot distinguish 5hmC from 5mC, so the methylation concurrence ratio is an underestimation of the real concurrence. This can be improved by applying oxidative bisulfite-seq, a technique that can discriminate between 5mC and 5hmC[40]. Additionally, the current study only associates methylation concurrence ratio with the colocalization of de novo methyltransferases DNMT3A/B and active demethylase TET. The maintenance methylation by DNMT1 and passive demethylation due to DNA replication are not discussed. In future work, a more comprehensive model is needed to incorporate all these factors.

Polycomb repressive complex 2 (PRC2) is responsible for methylating histone H3 on Lys27 (H3K27)[41], a crucial chromatin mark for gene silencing in early development and oncogenesis[42]. Both single-molecule[43] and genome-wide[44] analysis reveal that PRC2 depositing is not favored by DNA methylation. Consistent with these findings, the PRC2 catalytic subunit EZH2 (enhancer of Zeste homologue 2)[45], the stimulating subunit SUZ12 (suppressor of Zeste 12)[46], and H3K27 trimethylation (H3K27me3) are all negatively associated with the average methylation in promoters in human stem cells (Supplementary Fig. 17b). However, they are all positive indicators of methylation concurrence (Supplementary Fig. 17a). Bisulfite sequencing data in Ezh2 conditional knockout mice[47] also confirm that the removal of PRC2 would cause more regions to have higher average methylation but lower methylation concurrence (Supplementary Fig. 17c), suggesting that methylation concurrence can be promoted by PRC2 binding. Previous studies suggest that TET1 is associated with the repression of Polycomb targets[48,49], but have not detected any direct interaction between PRC2 and TET1. Our analysis indicates that methylation concurrence may serve as the missing link in the PRC2-TET1 association. Although the methylation concurrence corresponds with transcription silence, however, the current data are not sufficient to prove its mechanistic connection. Future works are needed to investigate its difference from the well-known Polycomb repression model.

Nucleosome positioning is essential for gene regulation by altering chromatin accessibility[50]. Nucleosome fuzziness measures the randomness of a nucleosome's position. Through reanalyzing human brain MNase-seq (micrococcal nuclease digestion with deep sequencing) data[51], we reveal that the methylation concurrence ratio is not associated with nucleosome fuzziness (Supplementary Fig. 18), suggesting that methylation concurrence regulates gene expression independent of nucleosome positioning.

Methylation canyons are poorly methylated and have negligible differences in the average methylation, so they serve as an ideal context to investigate the methylation concurrence. A previous study of mouse hematopoietic stem cells (HSCs) finds that some

canyons are active while the others are silent, and that such a difference is explainable by the enrichment of H3K4me3 or H3K27me3[33]. Thanks to our definition of methylation concurrence, the distinct activities of different canyons can be explained by methylation data alone, suggesting that methylation concurrence indicates chromatin accessibility. Besides, we find that the active canyon (aCanyon) target genes present a unique regulating model: negative correlation between gene expression and gene-body methylation (Fig. 4d). Although we have confirmed this observation by analyzing in vitro SELEX data of the 5mC preferences of TFs, it is worth noting that current model (Fig. 4g) can only explain the association between methylation change and binding of transcription activators but not repressors. In the future, further efforts are required to elucidate this mechanism.

With its different characteristics from the average methylation and methylation variation, the methylation concurrence highlights local methylation abnormality in the epigenetic landscape and will serve as a unique layer of methylation biology.

## Methods

**Quality control and reads alignments**. FastQC v0.11.7 (https://www.bioinformatics.babraham.ac.uk/projects/fastqc/) was used for general quality checks of sequencing reads in FASTQ files. Trim Galore v0.6.4 (https://github.com/FelixKrueger/TrimGalore) was used to trim the sequencing adaptor and remove low-quality bases. WGBS reads were aligned to human (hg19) or mouse (mm9) genome using BSMAP v2.90[52] with default parameters. RRBS reads were also aligned by BSMAP, while an extra option '-D C-CGG' was added to activate the RRBS mode[53]. The overlapping bases of two read mates were only counted once to avoid duplicate counting. RNA-seq reads were first mapped to human (hg19) genome by STAR v2.6.0c[54] with the option '–quantMode TranscriptomeSAM', and then gene expression level in FPKM (fragments per kilobase per million reads) was quantified using RSEM v1.3.1[55]. MNase-seq reads from the human brain were mapped to the human genome (hg19) by bowtie2 v2.2.7[56], then nucleosome binding positions and nucleosome fuzziness scores were called by DANPOS v2.2.2[57] with the 'dpos' command. The Hi-C reads from CD3 + T cells were mapped to the human genome (hg19) by bowtie2. Chromatin interactions under 2-kilobase resolution were then called using the 'analyzeHiC' module in Homer v4.8[58]. Samtools v0.1.19 (http://samtools.sourceforge.net/) was used to process reads alignment files.

**Genome annotation and genomic features**. The NCBI RefSeq gene annotation files (hg19 and mm9) were fetched using the Table Browser tool in the UCSC Genome Browser (http://genome.ucsc.edu/cgi-bin/hgTables). Gene promoter regions were defined as 1 kb upstream of TSS to 500 bp downstream of TSS. Promoters' CpG ratios were calculated by the following formula: (number of CpGs × number of bp)/(number of Cs × number of Gs)[25]. Then promoters were stratified into three groups according to their CpG ratios, i.e., high-CpG promoters (HCPs), intermediate-CpG promoters (ICPs), and low-CpG promoters (LCPs). Gene-body regions were defined as 500 bp downstream of TSS to the transcription termination site (TTS). CpG-island positions were fetched using the UCSC Table Browser, which was derived based on the published formula[59]. Enhancers and their target genes in CD3 + T cell were defined based on Hi-C data. First, the pairs of anchors, which have chromatin interaction were annotated to the genome. Then for each pair, if only one of them located in promoter or gene-body regions, it was assigned as an enhancer-target pair.

**The quantification of concurrence between active DNA methylation and demethylation**. The methylation concurrence events are represented by the unmethylated CpGs in partially methylated reads (red circles in Fig. 1a). Bisulfite sequencing reads were thus dissected into three categories of fragments, i.e., methylated fragments (consecutive solid circles in Fig. 1a), unmethylated fragments (consecutive blank circles in Fig. 1a), and methylation concurrence fragments (consecutive red circles in Fig. 1a). Thus, the methylation concurrence ratio of a particular genomic region was measured by the following equation.

$$\text{Methylation concurrence ratio} = \frac{\sum_{c=1}^{C} \omega_c}{\sum_{c=1}^{C} \omega_c + \sum_{m=1}^{M} \omega_m + \sum_{u=1}^{U} \omega_u} \quad (1)$$

$M$, $U$, and $C$ represent the numbers of methylated, unmethylated, and methylation-concurrence fragments, respectively. $\omega_m$, $\omega_u$, and $\omega_c$ are the weights for each fragment. They can be set as the CpG counts of each fragment (weighted) or 1 (unweighted) (Supplementary Fig. 1a). Comparing the weighted and unweighted ratios, we found that the weighted ratio (Fig. 2b; left panel) has slightly better correlations with gene expression than the unweighted ratio does (Supplementary Fig. 20a). Hence, the weighted methylation concurrence ratio was

used in this study. Note that the weighted methylation concurrence ratio is equivalent to the proportion of CpGs in methylation-concurrence fragments among all the CpGs. Therefore, the methylation concurrence ratio is also applicable to a single CpG site. Theoretically, methylation concurrence ratios vary between 0 and 1 (not equal to 1). In real data, however, as more than half of reads are either fully methylated or fully unmethylated (Supplementary Fig. 20b), methylation concurrence ratios of most regions are lower than 0.5 (Supplementary Fig. 20c). To reduce false positives, we only retained CpGs covered by at least four reads in the analysis. We integrated the calculation of the methylation concurrence ratio and the average methylation ratio in a single Python script available at https://github.com/JiejunShi/CAMDA.

**Quantifications of the average methylation**. Two measures of the average methylation are used in this study: traditional mean methylation and cellular heterogeneity-adjusted mean methylation (CHALM)[21]. The traditional mean methylation of a region is calculated as the proportion of methylated CpGs among all CpGs covered by reads. The CHALM value of a region is computed as $n_m/(n_m + n_u)$, where $n_m$ and $n_u$ are the counts of methylated reads and unmethylated reads, respectively.

**Visualization of the methylation concurrence ratio and the average methylation**. Methylation-concurrence ratios and average methylation measures at base-pair resolution were generated in the 'wiggle' format using our 'methylation concurrence' tool (https://github.com/JiejunShi/CAMDA). The 'wigToBigWig' program from UCSC binary utility directory (http://hgdownload.soe.ucsc.edu/admin/exe/) was used to transform 'wiggle' files to 'bigWig' files. For viewing specific gene loci, the 'bigWig' files were uploaded to 'custom tracks' in UCSC Genome Browser (http://genome.ucsc.edu/), with genome assembly of hg19 or mm9. The 'computeMatrix' module of deepTools v3.2.1[60] was used to extract the scores of interested regions into matrix format. Then the averaged profiles and heatmaps were visualized using 'plot' and 'image' functions in R.

**The colocalization analysis of DNMT3A1 and TET1**. The ChIP-seq reads density profiles of DNMT3A1 and TET1 in mouse ESCs were downloaded from the GEO database (GSE100951 and GSE100955). They have been corrected for input background using DANPOS. The reads densities in promoter regions were then extracted and transformed into log2 RPKM values by normalizing for promoter lengths and total reads counts. Hence, the promoter binding intensities of DNMT3A1 or TET1 were defined as the standardized values $\pi = \frac{x-\mu}{\sigma}$, where $\pi$ is the binding intensity, $x$ is the log2 RPKM, $\mu$ is the mean of $x$ across promoters, and $\sigma$ is the standard deviation of $x$ across promoters. The 'DNMT3A1-TET1 joint regulation score' ($\Pi$) of a promoter region is defined in the following equation.

$$\Pi = \frac{\pi^D - \pi^D_{\min}}{\pi^D_{\max} - \pi^D_{\min}} \cdot \frac{\pi^T - \pi^T_{\min}}{\pi^T_{\max} - \pi^T_{\min}}, \qquad (2)$$

where $\pi^D$ and $\pi^T$ represent the respective binding intensities of DNMT3A1 and TET1 in the promoter region; $\pi^D_{\min}$ and $\pi^T_{\min}$ indicate the minimum of $\pi^D$ and $\pi^T$ across promoters; $\pi^D_{\max}$ and $\pi^T_{\max}$ indicate the maximum of $\pi^D$ and $\pi^T$ across promoters. By definition, $\Pi$ is between 0 and 1.

**Quantifications of the methylation variation**. DNA methylation variation is quantified by the 'methylation heterogeneity'(MH) scores. In the MH algorithms, reads that cover fewer than 4 CpGs are excluded. As with the quantification of the methylation concurrence, only CpGs covered by at least four reads were included in the analysis to reduce false positive. The proportion of discordant reads (PDR) was calculated as previously described[12]. The PDR for a 4-CpG locus was the proportion of partially methylated reads among all reads, which cover the four adjacent CpGs. Epipolymorphism[13] for a 4-CpG locus was calculated as $1 - \sum(p_i^2)$, where $_{pi}$ is the frequency of all possible methylation patterns of this locus. Methylation entropy for a 4-CpG locus is measured by the equation of Shannon's entropy, $-\sum(p_i \ln p_i)$, where $_{pi}$ is the same as in Epipolymorphism. The entropy/Epipolymorphism/PDR of a region is calculated as the average value of all 4-CpG locus in that region. The entropy value of 4-CpG loci ranges from 0 to 4. We thus set a cutoff of 0.4 to define a notable entropy change. A comparison study[61] has implemented the above MH algorithms, and the codes are available on GitHub (https://github.com/MPIIComputationalEpigenetics/WSHScripts).

**Differentially methylated regions (DMRs) and differential methylation-concurrence regions (DMCRs)**. DMRs and DMCRs are identified by Metilene v0.2-8[62] in 'de novo' mode with the option of '-m 5' to get regions of at least 5 CpGs. For DMRs (hypermethylated and hypomethylated regions), '-d 0.4' is used to detect the regions with at least 40% average methylation difference. For DMCRs (concurrence-elevated and concurrence-depleted regions), because methylation concurrence scores of most regions are lower than 0.5 (Supplementary Fig. 20c), '-d 0.2' is applied to find regions with at least 20% methylation concurrence difference. The p value cutoff for both DMR and DMCR is 0.05.

**Gene set enrichment analysis (GSEA) and gene ontology (GO) analysis**. GSEA was applied by GSEA software v4.0.3[63] in the 'pre-ranked' mode with default parameters. The genes were decreasingly sorted by the promoter methylation concurrence ratio. All the functional terms are collected in GSEA Molecular Signatures Database (MsigDB, v7.1), except 'COSMIC Tumor Suppressors', 'COSMIC Oncogenes', and 'PANCAN Driver Genes'. 'COSMIC Tumor Suppressors' and 'COSMIC Oncogenes' were fetched from COSMIC Cancer Gene Census (CGC). 'PANCAN Driver Genes' were downloaded from a previous study[64], which integrated cancer drivers from 33 cancer types. The normalized enrichment scores (NES) and FDR values were calculated by GSEA software. The heatmap of NES was plotted using the 'image' function in R. Running enrichment scores were replotted using codes published on GitHub (https://github.com/PeeperLab/Rtoolbox/blob/master/R/ReplotGSEA.R). The GO analysis of methylation altered genes was conducted using the DAVID online tool (https://david.ncifcrf.gov/)[65]. Benjamini–Hochberg adjusted p values were calculated by DAVID. The functional enrichment of methylation canyon target genes was measured by two-tailed Fisher's Exact Test, and the p values were adjusted using the Benjamini–Hochberg method.

**Identifying differential methylated genes between tumor and normal samples**. The promoter differential methylation was defined by Metilene ('pre-defined regions' mode) if the methylation data contained replicates. The cutoff for significant methylation is p value < 0.005.

Because there is no replicate for WGBS data in the matched normal/tumor biopsy samples for UCEC and BRCA, the classical p-value-based hypothesis testing framework for identifying differential signals does not apply. To identify the statistically reliable difference, we used the local false discovery rate (local-fdr), a statistical criterion that assesses the credibility of individual discoveries under the Bayesian framework[29].

Let $d_j$ denote the difference between matched tumor and normal samples based on any methylation measure (e.g., average methylation) for gene $j$, $j = 1, \ldots, p$. Local-fdr assumes that genes come from two populations: differential and nondifferential. Let $p_0$ denote the prior probability that a gene is nondifferential. Let $f_0(d) = P(D = d|\text{nondifferential})$ and $f_1(d) = P(D = d|\text{differential})$ denote the conditional probability density of $D$ at $d$ given that $D$ comes from the nondifferential and the differential gene population, respectively. Thus, by Bayes' theorem, the posterior probability of a gene being nondifferential given its summary statistics is $P(\text{nondifferential}|D = d) = p_0 f_0(d)/f(d)$, where $f(d) := p_0 f_0(d) + (1 - p_0)f_1(d)$ is the marginal probability density of $D$. The local-fdr of gene $j$ is then defined as $\text{local}-fdr_j = \frac{f_0(d_j)}{f(d_j)}$. Because $p_0 \leq 1$, $\text{local}-fdr_j$ is an upper bound of $P(\text{nondifferential}|D = d_j)$. Notably, local-fdr differs from the traditional FDR in that FDR ensures the reliability of the identified differential genes as a set while local-fdr focuses on the reliability of each identified differential gene. To implement local-fdr, $f_0$ is assumed to be normal distributed and is estimated from middle-ranged $d_j$'s and $f$ is estimated nonparametrically from $\{d_1, \cdots, d_p\}$, as in the R package 'locfdr'.

To set the same threshold for both hyper and hypomethylated genes, we used the absolute differences rather than the signed differences and assumed that $f_0$ follows half-normal, a normal distribution with mean 0 and non-negative support. In the DNA methylation field, the conventional threshold for methylation difference is 10%[66]. To control this posterior probability, we selected genes whose methylation difference lower than 10% to estimate $f_0$. To make sure the data used for fitting $f_0$ looks like normal, we picked genes whose methylation difference reside in the middle one-third interval. We excluded the left one-third region to avoid high peaks close to 0, which deviate from the shape of a normal distribution; we also excluded the right one-third region to avoid fitting a mixture of differential and nondifferential genes. To estimate $f$, we used kernel density estimation. Given $\hat{f}_0$ and $\hat{f}$, the estimates of $f_0$ and $f$, we estimated local-fdr of gene $j$ by $\widehat{\text{local}-fdr}_j = \hat{f}_0(d_j)/\hat{f}(d_j)$. Finally, we call a differentially methylated gene if its estimated local-fdr is lower than 0.2, the cutoff suggested by the original local-fdr paper.

**Defining undermethylated regions (UMRs)**. UMRs were detected using the published method[33] based on the Hidden Markov model. To reduce false positives, we retained only CpGs covered by at least four reads. CpGs with an average methylation ratio lower than 10% were defined as undermethylated CpGs. UMRs were then identified as regions that include at least four consecutive undermethylated CpGs. The adjacent UMRs were merged into a single UMR if the average methylation ratio of the merged UMR was lower than 10% after merging.

**Determining the threshold for categorizing methylation canyons**. Methylation canyons are large (≥3.5 kb) and conserved UMRs[33,35,36]. UMRs longer than 3.5 kb were identified as canyons. Canyon target genes were defined as genes whose promoter or gene-body region is overlapped with canyons. To set the threshold for canyon categorization, we calculated the methylation concurrence ratios for 10,000 nonoverlapped random genomic regions to determine the distribution of genome backgrounds. 'fitdist' function in R package 'fitdistrplus' was then used to fit these to the classical distributions, including 'normal', 'log-normal', 'beta', 'gamma',

'uniform', 'exponential', and 'logistic' distributions (Supplementary Fig. 13a). The parameters for each distribution were estimated by 'maximum likelihood estimation'. By using the Cramér–von Mises criterion, we found the fitness to 'gamma' has the minimum distance with the distribution of genome background methylation concurrence ratios, which means 'gamma' distribution is the best fit (Supplementary Fig. 13b). This was also confirmed by the Q–Q plot (Supplementary Fig. 13c). Given the estimated parameters 'shape' and 'rate' from 'gamma' fitness, the threshold was determined as the 90th percentile of the background distribution, which aims to select the regions with significantly higher methylation concurrence ratios. Canyons were categorized based on this threshold. Canyons with methylation concurrence ratios higher than this threshold were assigned as 'pCanyons' (Polycomb canyons), while the remainder were designated as 'aCanyons' (active canyons) (Supplementary Fig. 13d).

**Locus-specific correlation between average methylation and gene expression**. The gene promoter and downstream regions (2 kb upstream to 10 kb downstream of TSS) were equally divided into 120 bins. The average methylation ratio was calculated for each bin, which is 100 bp in length. Aligning the TSS of different genes, all of the average methylation ratios were organized into a matrix, in which rows are genes and columns are bins. Then, Spearman's rank correlation coefficients were computed between the gene expression vector and each column of this matrix. The resulting vectors of correlation coefficients were visualized as bar plots.

**Enrichment of motifs with different 5mC preference**. The 5mC preferences of TF motifs were quantified by the SELEX method introduced by the previous study[38]. The motifs with preference values (termed as 'mCG enrichment' in ref. [38]) higher than 0 were assigned as 'methyl-plus' motifs, whose binding can be enhanced by 5mC. Motifs with preference value lower than 0 were defined as 'methyl-minus' motifs, whose binding is not favored by 5mC. Besides these two categories, there are also motifs with either multiple effects or little effect on 5mC. Removing these ambiguous motifs and motifs with identical sequences, we received 105 nonredundant motifs that are confidently assigned as 'methyl-minus' or 'methyl-plus'. The 'findMotifsGenome' module in Homer software was then used to call the motif positions in canyon sequences. The enrichment score of a particular motif was calculated as motif counts per kilobase (CPK) of the canyon. The fold enrichment of this motif was defined as the fold change between its CPK at aCanyons and CPK at pCanyons. Spearman's rank correlation between fold enrichment and 5mC preference was visualized in a scatter plot. The $p$ value was calculated by correlation test. A similar analysis was also performed on canyon gene promoters.

**Transcription factor binding difference between methylation canyon groups**. Positions of TF binding peaks determined in H1 human stem cells were downloaded from the Roadmap website. The percentages of TF-occupied aCanyons and pCanyons were then calculated based on whether the canyon overlaps with TF binding peaks. The binding difference was measured by the odds ratio between these two percentages, i.e., TF-occupied aCanyons (%)/TF-occupied pCanyons (%). TFs with odds ratios higher than 1 are more enriched in aCanyons, while the others are more enriched in pCanyons. A similar analysis was also performed on canyon gene promoters.

**Chromatin interaction at methylation canyons**. Chromatin interactions were defined by pairs of Hi-C anchors. The overlap between anchor pairs and canyons was detected using bedtools v2.25.0[67]. To exclude Canyon length bias, we also checked chromatin interactions in two control groups. Using the 'shuffleBed' function in bedtools, we generated a random genomic region for each Canyon whose length is equal to the Canyon. Then, the random regions whose lengths are equal to aCanyons were designated as the control group of aCanyons, while the random regions whose lengths are equal to pCanyons were designated as the control group of pCanyons. For each anchor pair, if both are located in the same canyon or random genomic region, the anchor pair is denoted 'self-interacting'. If only one of them is located in a canyon or random region, that anchor pair is denoted 'distant-interacting'.

**Reporting summary**. Further information on research design is available in the Nature Research Reporting Summary linked to this article.

## Data availability

All the public or controlled data used by this study are summarized in Supplementary Data 1. In total, 75 methylomes (WGBS and reduced representation bisulfite sequencing (RRBS)) were collected from the Roadmap Epigenomics project (http://www.roadmapepigenomics.org/), Encyclopedia of DNA Elements (ENCODE) project (https://www.encodeproject.org/), The Cancer Genome Atlas (TCGA) project (https://portal.gdc.cancer.gov/), Gene Expression Omnibus (GEO, https://www.ncbi.nlm.nih.gov/geo/), and DNA Data Bank of Japan (DDBJ, https://www.ddbj.nig.ac.jp) databases, including 69 human and 6 mouse datasets. All of them are publicly available except two matched normal-tumor pairs from TCGA, which need to apply for accession from GDC portal (https://portal.gdc.cancer.gov/). Ten RNA-sequencing

(RNA-seq) datasets were downloaded from Roadmap and TCGA, which are from the matched samples with methylomes. 69 ChIP-seq, 2 DNase-seq, 1 MNase-seq, and 1 Hi-C datasets were fetched from Roadmap or GEO. Source data for Fig. 3 and Fig. 4 are provided as Supplementary Data with the paper. Other data that support the findings of this study are available at https://github.com/JiejunShi/CAMDA/tree/master/paper-data.

## Code availability

The open-source software for methylation concurrence is freely available at https://github.com/JiejunShi/CAMDA.

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

## Acknowledgements

We thank members of the Li lab for helpful discussions. This work was supported by the US National Institutes of Health to W.L. (R01HG007538, R01CA193466, R01CA228140).

## Author contributions

J.S. and W.L. conceived and developed the outline of this research. J.S. wrote the tools and performed data analysis and method evaluations. J.X., Y.E.C., J.J.L., J.S.L., L.S., and Y.C. assisted with the manuscript. J.S. and W.L. wrote the paper.

## Competing interests

After completing the current studies at Baylor College of Medicine, J.X. became a full-time employee at Helio Health. W.L. is a consultant for Helio Health and ChosenMed. The remaining authors declare no competing interests.
