## [Peer Review File · Nature Communications]

The concurrence of DNA methylation and demethylation is associated with transcription regulationREVIEWER COMMENTS

Reviewer #1 (Remarks to the Author):

Shi et al. propose a new metric for evaluating read-level DNA methylation to improve resolution and prediction of transcriptional behavior as it relates to this epigenetic modification. Their metric, termed "the competition ratio" acts very similarly to other described approaches by prioritizing DNA fragments that have methylated and unmethylated CpGs in phase, but appears to be somewhat more sensitive than entropy and more precise than "discordance." They introduce this measurement in mouse embryonic stem cells to demonstrate its correlation with opposing DNMT3 and TET recruitment. Then, they apply it to a number of contexts where clear interpretation of DNA methylation data remains lacking, including of promoter silencing in cancer and for distinguishing between active transcription and repression by Polycomb in primary samples. Cumulatively, I find this new measurement interesting and believe the authors are seeking to address an unmet need in the field. However, in its current form I feel the manuscript is somewhat shallow, using only one or a handful of samples to highlight different points without adequately generalizing their findings. Similarly, I feel that they should be more rigorous in how their lens relates to currently published measurements, what distinguishes the "competition ratio," as well as how they apply or interpret their current results.

Major issues:

1. Overreliance on single cell types for different statements: Currently the paper elects to cherry pick select public samples to address key points, making it difficult to appreciate the extent to which their claims are generalizable. Figure 1 uses mESCs, Figure 2 and 4 CD3+ T cells, and Figure 2 in one TCGA patient-matched normal/tumor biopsy.

Figure 1 justifies the nomenclature of "competition" by examining public data of TET and DNMT3 enrichment in wild type ESCs and validate it by examining TET and DNMT3 KO mouse embryonic stem cells (mESCs). However, mESCs show substantial global turnover that appears very unique to this cell type. How does the competition metric change for differentially methylated regions discovered in TET and DNMT3 KO human Embryonic Stem Cells (Charlton et al., Nature Genetics 2020, Ref. 47) or in the variety of hematopoiesis-specific knockout models described by the Goodell lab?

Similarly, the association with promoter methylation during tumorigenesis (Figure 3) and differences between active and polycomb Canyons both seem well positioned for meta-analysis across the many samples for which the authors have applied this metric. How tissue-specific versus shared are aCanyon and pCanyon designations? Does the high enrichment for the competition ratio hold for pCanyons in any given cell type but change to the alternative trends in contexts when it is expressed? Even if the authors cannot validate their a/pCanyon nomenclature using the full suite of modifications available in CD3+ T cells, they could still use that cell type to justify examining others where DNA methylation and expression data are available. There are probably many cell types for which orthogonal H3K27me3 or H3K4me3 data are also available.

For the methylation canyon section, I would have expected additional description using either the public TET / DNMT3 KO data or in patient-matched normal/tumor biopsies. How does the competition ratio change across differentially methylated canyons compared to the change in mean methylation, and what are the author's biological interpretations? These regions are frequently found within large partially methylated domains, so the expectation is that they will be increasingly discordant. However, "competition" is already high in polycomb-dominated regions, where DNA methylation values are low. Is the increase in mean methylation more substantial than the increase in competition ratio?

2. The mechanistic interpretation of the score: I am uncomfortable with the name of this metric as it implies a mechanistic link that is not strongly validated. The competition ratio represents a read-level methylation estimate that is possibly more precise than PDR (which assigns any discordant read the same value). Overall, it seems very similar to entropy, though possibly better bounded (entropy will scale differently according to the number of CpGs in each string). Really the only justification of a direct mechanistic link is the correlation with ChIP-seq signal intensity calculated

across promoter sequences. This data is taken from a paper that shows that, in WT, these enzymes do not overlap so much as Dnmt3a1 flanks a subset of regions bound by Tet1 (as visualized in Figure 1e). In using this name, the authors risk interpretation from readers that repression is in fact mediated by the dual actions of these competing enzymes. I feel like this is particularly dangerous given the enrichment for this measurement across Polycomb-regulated territories, where genes can still be transcriptionally induced and where the repressive mechanism is clear. Competition is also high at tumor suppressor genes in cancer, where presumably H3K27me3 has been depleted and the gene is more terminally/irreversibly silenced.

Phrases like “We quantify the competition between DNA methylation and demethylation” should be changed to make clear that this method only infers or estimates turnover, but is not directly mechanistic. Possibly they should also include a statement somewhere in the conclusions about how, although this measurement corresponds with being in a transcriptionally silent state, it cannot apparently distinguish between repressive modes or mechanisms (unless they can show better evidence that it can).

3. Comparison between the competition ratio and other metrics for measuring DNA methylation: the comparison between the competition ratio and other metrics strikes me as relatively shallow and sometimes possibly unfair. For example, statements about the poor correlation of mean promoter methylation appears to be calculated using Spearman correlation. However, DNA methylation is highly bimodal, the majority of promoters are either highly unmethylated or highly methylated, such that the majority of the data range is low density. If a Pearson correlation was calculated between these metrics on the unbinned/discrete data, I imagine we will see a stronger negative correlation between these other methylation metrics (including mean) and gene expression.

Similarly, I feel that binning the data into 25 subsets across their range is somewhat misleading as it shifts the representation of genes for each metric in each bin. The competition ratio could appear to be a better predictor of gene expression purely because it is more sensitive at these lower values. We also see evidence for this in Figure 2a, when genes are binned by their expression value (instead of methylation readout). Here, the difference between the lowest expression quartile and the other three is virtually indistinguishable (in magnitude) between “Competition,” Average, and Variation (Entropy). At the very least, the plots in Figure 2b and Extended Data Figures 2, 3, 4 and 6 should include information about the representation of elements present in each bin.

From this perspective, one may expect mean methylation to be more predictive for some promoters than competition, and vice versa. Genes with hypermethylated promoters are expected to be highly repressed, yet these would also show low competition scores. More work should be done to compare these measurements to one another or to combine them, as opposed to showing their independent performance on gene expression. For example, I would like to see scatterplots that show the competition ratio on one axis and other metrics (including mean) on the other, similar to ED Figure 16. Conceivably, one could add the third dimension of gene expression (or at least expression quantile) using color.

4. Deeper understanding of how competition distinguishes or identifies differentially regulated regions in cancer: The authors show that the competition ratio can identify dynamically regulated regions between normal and cancer samples in two matched tissue/tumor biopsy samples. However, the difference between these regions and those that can be identified by mean methylation is limited to gene set enrichment and overlap with a limited set of predefined tumor suppressor genes. In my opinion, the relevance of these competition-only targets is still largely speculative. Based upon visual inspection of the P1, P2, and P3 subsets, it seems like mean methylation is still quite highly associated with gene repression (as indicated by the tumor boxplots going to near zero). In contrast, the competition ratio-only subset (P1) appears to be more false positive prone (the overall expression of these genes in tumor remains higher than for P2 and P3). Similarly, the change in the competition ratio appears to be more minimal in P1 than in P3. It very much appears that competition may be more sensitive to subtle changes in DNMT activity, but therefor requires a more careful statistical delineation between significant and non-significant.

Also, given the prior claim that the competition score shows a greater correlation with gene expression within a sample (as discussed above), I don't see a discrete claim here about the precision of this score to predict changes in gene expression instead of being merely associated. Again, the boxplots in Figure 3c and ED Figure 8c show general trends that support some overlap between differential gene expression, competition and/or mean promoter methylation. However, what is the overlap between repressed genes and promoter sequences that are differential according to these two scores? What is the relationship between change in competition and change in expression and how does this differ from mean? One could imagine showing this with scatter plots showing the difference in promoter methylation and gene expression between cancer and normal.

5. Description and analysis of high and low competition canyons. To me, the current description and findings associated with the competition ratio at Polycomb regulated (p) versus active (a) canyons is largely not novel and distorted by a few missing variables/descriptions. For example, large domains of unmethylated DNA are apparent for Polycomb regulated targets, whereas smaller canyons are associated with CpG islands at highly transcribed genes. This is born out by the author's analysis of public CD⁺ T cell data: aCanyons are highly enriched for euchromatic chromatin modifications associated with transcription (H3K4me3) and are highly transcribed. It's therefore not surprising that they would show low "competition" (discordance) within the gene body, as the gene bodies of actively transcribed genes are in general very highly methylated. The negative correlation between gene expression and competition in Figure 4d is very likely explained by these gene bodies being very highly methylated, and cannot be captured by the competition score. At the very least, the authors should include data showing the mean methylation tracks for aCanyon and pCanyon genes to give the reader a frame of reference.

I suspect that pCanyons are going to be substantially larger than aCanyons, consistent with an extended Polycomb footprint and the fact that valley prediction will likely truncate abruptly at the boundary of a transcribed gene body (similarly, the position of genes relative to the valley boundary may be more obvious). These additional genomic features can also be represented using composite style plots such as in Figure 4a.

Normalized figures such as those associated with Figure 4a would be misleading if the length of sequences between "left" and "right" are structurally different. Similarly, differences in length would bias the results described in ED Figure 13, as pCanyons will have more bases from which to capture an interaction. If this is the case, the "random" background would also have to be adjusted for the unique features of each subgroup.

This issue may also hold for the analysis of TF binding preferences. If the background of pCanyons and aCanyons is structurally different (ie. if aCanyons are more focally restricted to the TSS of actively transcribed genes whereas pCanyons are larger and less well positioned with respect to the TSS of the relevant gene), you may expect differential enrichment of generic transcriptional activators, such as TAF1 and TBP, purely as a bioinformatic artifact. I would be curious if these TF enrichments hold if motif analysis is fixed to the TSS (or CpG islands) of genes that reside within Canyons, as opposed to scanned agnostically across the entire domains (which I believe from the Methods is what's happening).

Minor comments:

1. The statement in the introduction: "For example, the DNMT3A and TET2 double-knockout mice show worse survival than single-knockout counterparts," should qualify that this is in conditional knockout specific to the hematopoietic system.
2. I am curious how the TET and DNMT3A KOs impact other read-level methylation estimates beyond mean and the competition ratio. As I understand it, the competition ratio is similar to other metrics such as PDR, Epipolymorphism, or Entropy, in that it should be higher when read-level methylation is maximally entropic and lower as it approaches fully unmethylated or methylated states. One may expect similar trends for these metrics: while the mean methylation direction changes in agreement with the enzyme class deleted, these other metrics may show trends that

resemble the competition ratio.

3. The schematic in Figure 1a does an adequate job of describing the nature of data that informs DNMT3 vs. TET competition, but does not relate how these data are used to calculate the score itself. More detail should be provided to show how simulated data such as this relates to the unweighted and weighted competition scores. For inspiration, the authors could consider Figure 2 from Sherer et. al Nucleic Acids Research 2020 (Ref. 57) or Figure 2 in Guo et. al Nature Genetics 2017 (Ref. 14).

4. What do the error bars represent in plots Figure 2b, and ED Figures 1, 2, 3, 4, and 6?

5. Could the authors provide a more rigorous explanation regarding why the competition ratio is structurally different than other variation metrics for distinguishing aCanyons and pCanyons? Qualitatively, I agree that it doesn't preserve transition peaks around the boundaries of pCanyons, but really these other metrics also indicate that canyon boundaries don't differ between classes, so I'm not clear on the added value (if you plotted the difference between aCanyons and pCanyons, I'm not sure competition would look much different).

Reviewer #2 (Remarks to the Author):

Shi and colleagues describe a new DNA methylation metric termed 'methylation competition' that appears to be better correlated with gene expression, silencing of tumor suppressor genes and segregates DNA methylation canyons into active and polycomb repressed canyons (aCanyons and pCanyons). Overall the manuscript has several interesting aspects but can be further improved in clarity and presentation.

Specific comments:

There is not enough evidence provided that methylation competition is involved in tumorigenesis as the title might suggest. The metric might explain some gene expression observations better but that is something else.

Similar in the abstract, the statement 'during tumorigenesis' and 'important for... tumorigenesis' would require more data than comparing the presented cancer and normal data. 'Largely distinct' is a rather vague ending for the abstract.

The metric 'methylation competition' is central to the paper but not well explained or introduced. Figure 1a explains the hypothetical bisulfites sequencing reads but should be expanded and refined to actually introduce the metric.

The text mentions (line 91) consecutive C, but that is not well explained.

It also ignores the competition that occurs on the filled circles that are the result of 5hmC (which will be called as mC in the data). Especially in the TET targeted regions one would expect this to be not insignificant.

It also doesn't account or consider other means of creating the partial methylation on fragments, such as Dnmt1s failure to maintain all methylation post replication or other factors that could interfere with Dnmt1 activity.

The difference to the other variation metrics could be explained better. As the authors state they are distinct in some aspects, but other behaviors are rather similar. This is related to the above point in the abstract.

The authors use mESCs that express also high levels of Dnmt3b and other TETs, however it is never explained why this is just focused on the Dnmt3a and TET1 competition.

Line 143 the knockouts are described but this is very confusing as it states Tet knockout and ONLY the legend clarifies that this a Tet1, 2, 3 triple knockout.

ZBTB16 is used as an example for a TSG, but no details (methylation data) for its promoter are provided. It's not clear which data support the strong conclusion 'that methylation competition at its promoter indicate a novel mechanism for its tumor suppressing role.'

Line 274, H3K27me3 is a histone modification, so bound by H3K27me3 is incorrect and should say 'enriched'. Also line 372.

1f,g: Add CGIs to the tracks

1f: It would be helpful to see the read level data for a regions like this.

1g: The average methylation shows a smaller peak in the unmethylated canyon, that should lead to some higher variation but doesn't show up in the track below.

1h: Dnmt3a KO in ESC should retain most methylation given high levels of Dnmt3b. More details should be provided on the knockouts and KO data.

1i: TET ko are quite similar in depleted and elevated regions. These should be triple knockout (see comment above about the confusing nomenclature that needs to be fixed in text and the Figure panel) and one should expect all the C cytosines to turn into M if one follows the authors logic, so it is surprising to see so many competition elevated regions.

Most other figures have few panels and lots of important points are in the supplement, this could be arranged more effectively. In turn, panels such as 2b don't have to be in the main Figures.

Reviewer #3 (Remarks to the Author):

The authors introduce a novel concept of methylation competition in an elegant way to identify regions that are bound by both methylation & demethylation agents within the same cell using bisulfite sequencing data.

The analysis on correlation with gene expression could benefit from more detail, how the results in Figure 2 are reported. Is this from all cells, from how many donors, etc. and across how many CpG sites/genes?

Sequencing depth: It is not clear why the authors would expect the relationship to be correlated with sequencing depth? Isn't this an artefact?

Tumor suppressor genes:

What is the interpretation that the three groups of genes in Figure 3a (P1, P2, P3) are enriched in different pathways? Any thoughts on biological interpretation of this finding?

Finally, can do authors provide more interpretation what the methylation competition metric's biological interpretation is? Any thoughts on what this all means?

Summary

First, we appreciate the perceived excitement among the reviewers as they clearly acknowledged the **significance and novelty** of our findings. For example:

- “Cumulatively, I find **this new measurement interesting** and believe **the authors are seeking to address an unmet need in the field.**” (Reviewer #1)
- “... the manuscript has **several interesting aspects** ...” (Reviewer #2)
- “The authors introduce **a novel concept of methylation competition in an elegant way to identify regions that are bound by both methylation & demethylation agents** ...” (Reviewer #3)

Second, all of the reviewers appeared to have read our manuscript carefully and have made many constructive suggestions. The major concerns include: (1) Further testing our new metric in other cell types (Reviewer #1); (2) Providing additional biological insights into the gene groups identified by the new metric (Reviewer #1 and #3); (3) Improving the comparison with other known metrics (Reviewer #1 and #2). Therefore, we have now performed the following additional analyses (**25 new figures** and **5 review figures**), which together significantly strengthen our findings:

- (1) Improving the explanation of our new metric
 - We change the new metric’s name from ‘methylation competition’ to ‘**methylation concurrence**’, which means the concurrence of DNA methylation and demethylation.
 - We elucidate how to calculate the methylation concurrence ratio in (**New Figure 1a**).
 - We add **New Figure S1** to compare different methylation metrics.
- (2) Generalizing our findings in more cell types
 - We further validate the effect of *DNMT3* and/or *TET* knockout on methylation concurrence in human embryonic stem cells. (**New Figure S5**)
 - We further confirm that methylation concurrence is also correlated with the co-localization of other DNMT3 and TET enzymes, such as DNMT3B and TET2 (**New Figure S4**).
 - We analyze the aCanyons and pCanyons in human embryonic stem cells (H1) and we confirm their distinct chromatin activity using all the available histone marks (**New Figure S14**).
- (3) Providing further biological insights into the gene groups identified by the new metric
 - We compare the aCanyons/pCanyons in H1 stem cells with those in CD3+ T-cell and we find that the changes of target gene expression are associated with aCanyons/pCanyons. (**Review Figure 1**)
 - We add two control groups, i.e., randomly selected highly expressed genes and randomly selected lowly expressed genes (**New Figure 4b, 4c, 4d**).
- (4) Improving the comparison with other known DNA methylation metrics
 - For the correlation with gene expression, all the “binning/grouping” style plots are replaced with the scatter plots with the unbinned raw data points (**New Figure 2b, New Figure S6b, New Figure S7, New Figure S9, and New Figure S19a**).
 - For the correlation with gene expression, we add Pearson correlation using the unbinned raw data (**New Figure 2b, New Figure S6b, New Figure S7, New Figure S9, and New Figure S19a**).
 - ‘Colored’ scatter plots to compare the gene expression correlation of different methylation metrics simultaneously (**Review Figure 3**).
 - We add the correlation between gene expression change and methylation difference between tumor and normal (**New Figure S8**).

- We apply the local false discovery rate (local-fdr), a statistical criterion that assesses the credibility of individual discoveries under the Bayesian framework, to identify the reliable methylation difference between TCGA normal and tumor samples (**New Figure 3a**, **New Figure S11b**, and **New Figure S12a**).
- We add the methylation variation scores in *Dnmt3* and *Tet* knockout mouse embryonic stem cells (mESCs) (**Review Figure 4**).
- We check the difference between pCanyons and aCanyons for the different methylation metrics (**New Figure S15**).

(5) Other improvements

- We add the methylation alteration due to *Tet* and *Dnmt3* knockout in mESCs (**Review Figure 2**).
- We add the length difference between aCanyons and pCanyons (**New Figure S16b**). To exclude such bias, we add two control groups of random regions for the chromatin interaction analysis (**New Figure S16d**). We also analyze the TF enrichment at Canyon gene promoters (**New Figure S16c** and **S16e**).
- We add the bisulfite-seq reads level data of example genes (**New Figure S3** and **New Figure S11a**) and regions (**Review Figure 5**).
- We remove vague statements which lack sufficient evidence and clarify the limitations of the current study in the Discussion.

In light of these substantial improvements, we are optimistic that all of the reviewers' concerns have been satisfactorily addressed. Please see below for a detailed **point-by-point response** (the reviewers' original comments are in blue).

Response to Reviewer #1:

Reviewer #1: Shi et al. propose a new metric for evaluating read-level DNA methylation to improve resolution and prediction of transcriptional behavior as it relates to this epigenetic modification. Their metric, termed "the competition ratio" acts very similarly to other described approaches by prioritizing DNA fragments that have methylated and unmethylated CpGs in phase, but appears to be somewhat more sensitive than entropy and more precise than "discordance." They introduce this measurement in mouse embryonic stem cells to demonstrate its correlation with opposing DNMT3 and TET recruitment. Then, they apply it to a number of contexts where clear interpretation of DNA methylation data remains lacking, including of promoter silencing in cancer and for distinguishing between active transcription and repression by Polycomb in primary samples. Cumulatively, I find this new measurement interesting and believe the authors are seeking to address an unmet need in the field.

Response: We thank the reviewer for acknowledging the significance and novelty of our findings.

However, in its current form I feel the manuscript is somewhat shallow, using only one or a handful of samples to highlight different points without adequately generalizing their findings. Similarly, I feel that they should be more rigorous in how their lens relates to currently published measurements, what distinguishes the "competition ratio," as well as how they apply or interpret their current results.

Major issues:

1. Overreliance on single cell types for different statements: Currently the paper elects to

cherry pick select public samples to address key points, making it difficult to appreciate the extent to which their claims are generalizable. Figure 1 uses mESCs, Figure 2 and 4 CD3+ T cells, and Figure 2 in one TCGA patient-matched normal/tumor biopsy.

Response: We thank the reviewer for the suggestions. We have included more cell types to test our new metric. Please see the details under specific comments below.

Figure 1 justifies the nomenclature of “competition” by examining public data of TET and DNMT3 enrichment in wild type ESCs and validate it by examining TET and DNMT3 KO mouse embryonic stem cells (mESCs). However, mESCs show substantial global turnover that appears very unique to this cell type. How does the competition metric change for differentially methylated regions discovered in TET and DNMT3 KO human Embryonic Stem Cells (Charlton et al., Nature Genetics 2020, Ref. 47) or in the variety of hematopoiesis-specific knockout models described by the Goodell lab?

Response: As suggested by the reviewer, we have added the methylation data in *TET* and/or *DNMT3* knockout human embryonic stem cells(Charlton et al., 2020). As expected, *DNMT3A* and *DNMT3B* double knockout (DKO) leads to a substantial decrease of average methylation, while *TET1*, *TET2*, *TET3* triple knockout (TKO) leads to hypermethylation. And the *DNMT3A*, *DNMT3B*, *TET1*, *TET2*, *TET3* pentuple knockout (PKO) sample shows a relatively “balanced” change of average methylation (**new Figure S5a**). In line with our previous findings in mESCs (Figure 1i), there are more depleted than elevated regions of methylation concurrence in DKO. The methylation concurrence in the TKO sample doesn’t show a significant trend, which also agrees with the mouse data (Figure 1i). Intriguingly, the knockout of both enzyme families (PKO) leads to a dominant trend of depleted concurrence (**New Figure S5b**). Altogether, these new data further justify the definition of methylation concurrence.

New Figure S5. DNA methylation alterations in *DNMT3* and/or *TET* knockout human embryonic stem cells (HUES8). (a) Average methylation altered regions in *DNMT3A* and *DNMT3B* double KO(DKO) HUES8, *TET1*, *TET2*, *TET3* triple KO(TKO) HUES8, and *DNMT3A*, *DNMT3B*, *TET1*, *TET2*, *TET3* pentuple KO(PKO) HUES8. (b) Methylation concurrence altered regions in DKO, TKO, and PKO HUES8.

Similarly, the association with promoter methylation during tumorigenesis (Figure 3) and differences between active and polycomb Canyons both seem well positioned for meta-analysis across the many samples for which the authors have applied this metric. How tissue-specific versus shared are aCanyon and pCanyon designations? Does the high enrichment for the competition ratio hold for pCanyons in any given cell type but change to the alternative trends in contexts when it is expressed?

Response: We thank the reviewer for the suggestions. To check how the active Canyons and Polycomb Canyons share between different samples, we compared the Canyons in human

embryonic stem cell (H1) with those in CD3+ T-cell. As indicated in **Review Figure 1a**, 137 pCanyon genes (T1 group) and 409 aCanyon genes (T2 group) are shared between the two cell types. In addition, 191 aCanyon genes in H1 become pCanyon genes in T-cells (T3 group), and 80 pCanyon genes become aCanyon genes (T4 group). The changes of methylation concurrence level between the two samples agree with their Canyon designations (**Review Figure 1b**). Due to the limited gene number, the gene expression and H3K27me3 change of the T4 group are not significant (**Review Figure 1c and 1d**). But the regulating patterns of all four groups are correlated with the methylation concurrence change. For example, the expression of the T3 group is lower in T-cell, which can be speculated from higher methylation concurrence and H3K27me3. In contrast, the expression of the T4 group is higher in T-cell, which agrees with their lower methylation concurrence and H3K27me3 levels.

Review Figure 1. Methylation Canyon targeted genes in H1 human embryonic stem cell and/or CD3+ T-cell. (a) Canyon genes overlap between stem cell and T-cell. The alterations of (b) methylation concurrence, (c) Gene expression, and (d) promoter H3K27me3 enrichment of T1, T2, T3, and T4 genes defined in (a).

Even if the authors cannot validate their a/pCanyon nomenclature using the full suite of modifications available in CD3+ T cells, they could still use that cell type to justify examining others where DNA methylation and expression data are available. There are probably many cell types for which orthogonal H3K27me3 or H3K4me3 data are also available.

Response: Thanks for the suggestions. For CD3+ T-cell, there are five histone modifications available in the “Epigenome Roadmap” project. Now we have added the other three histone marks (H3K27ac, H3K36me3, and H3K4me1) into the original Figure 4a. As shown in **New Figure 4a**, the active markers H3K27ac and H3K36me3 are higher in aCanyons. In contrast, the enhancer marker H3K4me1 doesn’t show a significant difference between aCanyons and pCanyons.

In addition to CD3+ T-cell, we also validated the chromatin activities of Canyons in H1 human ESC, in which 28 types of histone modifications/variants are available. As indicated in **New Figure S14**, active markers (H3K4me3, H3K27ac, H2A.Z, H3K9ac, etc.) are more enriched in aCanyons. In contrast, the repressive markers (H3K27me3, H3K9me3, etc.) are more enriched in pCanyons.

New Fig 4a. Average profiles of methylation concurrence, average methylation, H3K4me3, H3K27me3, DNase I hypersensitive sites, H3K27ac, H3K36me3, and H3K4me1 in methylation canyons in CD3+ T-cell. 'aCanyons' (red) are low-concurrence canyons. 'pCanyons' (blue) are high-concurrence canyons. The X-axis indicates the distance to canyon borders.

New Figure S14. Average profiles of methylation concurrence, average methylation, and 28 histone modifications/variants in methylation canyons in H1 stem cell. ‘aCanyons’ (red) are low-concurrence canyons. ‘pCanyons’ (blue) are high-concurrence canyons. The X-axis indicates the distance to canyon borders.

For the methylation canyon section, I would have expected additional description using either the public TET / DNMT3 KO data or in patient-matched normal/tumor biopsies. How does the competition ratio change across differentially methylated canyons compared to the change in mean methylation, and what are the author’s biological interpretations? These regions are frequently found within large partially methylated domains, so the expectation is that they will be increasingly discordant. However, “competition” is already high in polycomb-dominated regions, where DNA methylation values are low. Is the increase in mean methylation more substantial than the increase in competition ratio?

Response: Thanks for the suggestions. To gain insights into how the methylation is altered at Canyon regions, we called the methylation Canyons in wild-type (WT) mESCs and then checked their average methylation and methylation concurrence in WT, *Tet* TKO, and *Dnmt3* DKO mESCs. As shown in **Review Figure 2**, average methylation has similar patterns in both aCanyons and pCanyons, i.e., average methylation is decreased in *Dnmt3* DKO and increased in *Tet* TKO. In contrast, methylation concurrence is decreased in both *Dnmt3* DKO and *Tet* TKO. The alteration in *Dnmt3* DKO is more substantial than that in *Tet* TKO, which also agrees with previous analysis in Figure 1h and 1i.

Review Figure 2. The methylation alteration at Canyon regions due to *Tet* or *Dnmt3* knockout in mouse ES cells.

2. The mechanistic interpretation of the score: I am uncomfortable with the name of this metric as it implies a mechanistic link that is not strongly validated. The competition ratio represents a read-level methylation estimate that is possibly more precise than PDR (which assigns any discordant read the same value). Overall, it seems very similar to entropy, though possibly better bounded (entropy will scale differently according to the number of CpGs in each string). Really the only justification of a direct mechanistic link is the correlation with ChIP-seq signal intensity calculated across promoter sequences. This data is taken from a paper that shows that, in WT, these enzymes do not overlap so much as *Dnmt3a1* flanks a subset of regions bound by *Tet1* (as visualized in Figure 1e). In using this name, the authors risk interpretation from readers that repression is in fact mediated by the dual actions of these competing enzymes.

Response: Thanks for the suggestions. We totally agree that our new metric is not a mechanistic measure. Therefore, we decide to change its name to '**methylation concurrence**', which is the short term of 'the concurrence of DNA methylation and demethylation'. Methylation concurrence indicates that DNA methylation and demethylation occur simultaneously.

I feel like this is particularly dangerous given the enrichment for this measurement across Polycomb-regulated territories, where genes can still be transcriptionally induced and where the repressive mechanism is clear. Competition is also high at tumor suppressor genes in cancer, where presumably H3K27me3 has been depleted and the gene is more terminally/irreversibly silenced. Phrases like "We quantify the competition between DNA methylation and demethylation" should be changed to make clear that this method only infers or estimates turnover, but is not directly mechanistic. Possibly they should also include a statement somewhere in the conclusions about how, although this measurement corresponds with being in a transcriptionally silent state, it cannot apparently distinguish between repressive modes or mechanisms (unless they can show better evidence that it can).

Response: Thanks for the suggestions. Although the new metric corresponds with transcription silence, we agree that at present, we don't have enough evidence to prove its mechanistic connection. Although the high methylation concurrence genes have some overlap with Polycomb targets, further explorations are needed for the underlying mechanisms. In the Discussion, we clarify this limitation of the current study as *'Although the methylation concurrence corresponds with transcription silence, the current data are not sufficient to prove its mechanistic connection. Future works are needed to investigate its difference from the well-known Polycomb repression model.'*

3. Comparison between the competition ratio and other metrics for measuring DNA methylation: the comparison between the competition ratio and other metrics strikes me as relatively shallow and sometimes possibly unfair. For example, statements about the poor correlation of mean promoter methylation appears to be calculated using Spearman correlation. However, DNA methylation is highly bimodal, the majority of promoters are either highly unmethylated or highly methylated, such that the majority of the data range is low density. If a Pearson correlation was calculated between these metrics on the unbinned/discrete data, I imagine we will see a stronger negative correlation between these other methylation metrics (including mean) and gene expression.

Response: Thanks for the suggestions. First, we want to clarify that we used **unbinned raw data** to calculate the correlation between gene expression and methylation metrics in our original figures. The binning/grouping was only used to visualize the data. To avoid misunderstanding, we have replaced all the "binning/grouping" style plots with the scatter plots with the unbinned raw data points. This includes **New Figure 2b, New Figure S6b, New Figure S7, New Figure S9, and New Figure S19a**. As suggested by the reviewer, for the correlation with gene expression, we also calculated Pearson correlation using the unbinned raw data. As shown in **New Figure 2b**, both Pearson and Spearman correlation coefficients indicate that methylation concurrence better correlates with gene expression.

New Figure 2b. The promoter/gene-body/enhancer methylation concurrence ratios (1st column) are strongly negatively correlated with gene expression level in CD3+ T-cells, and this correlation is stronger than that of the average methylation (2nd column) and the methylation variation (3rd column). Promoter regions are from 1kb upstream to 500bp downstream of TSS. Gene-body regions are from 500bp downstream of TSS to TTS. Enhancer regions are defined based on chromatin interactions validated by Hi-C data (see Methods). Spearman's rank correlation and Pearson's correlation were calculated. P-values were calculated by the two-tailed correlation test. LOWESS lines were plotted to describe the relationships between variables (indicated by red curves).

New Figure S6b. The promoter CHALM scores (1st column) are negatively associated with gene expression level in CD3+ T-cell. But the correlation of gene-body (2nd column) and enhancer (3rd column) CHALM scores are very low. Promoter regions are from 1kb upstream to 500bp downstream of TSS. Gene-body regions are from 500bp downstream of TSS to TTS. Enhancer regions are defined based on chromatin interactions validated by Hi-C data (see Methods). Spearman's rank correlation and Pearson's correlation were calculated. P-values were calculated by the two-tailed correlation test. LOWESS lines were plotted to describe the relationships between variables (indicated by red curves).

New Figure S7. The methylation concurrence ratio is consistently associated with gene expression in various samples. The correlation between gene expression and promoter

methylation concurrence (1st column), promoter average methylation (2nd column), promoter methylation variation (3rd column), gene-body methylation concurrence (4th column), gene-body average methylation (5th column), gene-body methylation variation (6th column) in 4 different samples, i.e., CD14+ primary cells (1st row), CD56+ primary cells (2nd row), fetal thymus tissue (3rd row), and adult thymus tissue (4th row). Spearman's rank correlation and Pearson's correlation were calculated. P-values were calculated by the two-tailed correlation test. LOWESS lines were plotted to describe the relationships between variables (indicated by red curves).

New Figure S9b. The correlations between methylation quantitative measures and gene expression in three gene groups with different CpG densities at promoter.

Scatter plots show the correlation of HCP (1st row), ICP (2nd row), and LCP (3rd row) gene expression versus promoter methylation concurrence (1st column), average methylation (2nd column), and methylation variation (entropy, 3rd column), in CD3+ T-cell. Spearman's rank correlation and Pearson's correlation were calculated. P-values were calculated by the two-tailed correlation test. LOWESS lines were plotted to describe the relationships between variables (indicated by red curves).

New Figure S19a. The correlations between unweighted methylation concurrence ratios and gene expression.

The methylation concurrence ratios in CD3+ T-cells were calculated with $\omega_m = 1$, $\omega_u = 1$, and $\omega_c = 1$. Scatter plots show the correlation between gene expression and methylation concurrence ratios on promoter (left), gene-body (middle), and enhancer (right). Spearman's rank correlation and Pearson's correlation were calculated. P-values were calculated by the two-tailed correlation test. LOWESS lines were plotted to describe the relationships between variables (indicated by red curves).

Similarly, I feel that binning the data into 25 subsets across their range is somewhat misleading as it shifts the representation of genes for each metric in each bin. The competition ratio could appear to be a better predictor of gene expression purely because it is more sensitive at these lower values. We also see evidence for this in Figure 2a, when genes are binned by their expression value (instead of methylation readout). Here, the difference between the lowest expression quartile and the other three is virtually indistinguishable (in magnitude) between "Competition," Average, and Variation (Entropy). At the very least, the plots in Figure 2b and Extended Data Figures 2, 3, 4 and 6 should include information about the representation of elements present in each bin.

Response: Thanks for the suggestions. We would like to clarify again that correlation was calculated using **unbinned raw data** in our original figures. The binning/grouping was only used to visualize data. We have replaced all the "binning/grouping" style plots with the scatter plots the unbinned raw data points.

From this perspective, one may expect mean methylation to be more predictive for some promoters than competition, and vice versa. Genes with hypermethylated promoters are expected to be highly repressed, yet these would also show low competition scores. More work should be done to compare these measurements to one another or to combine them, as opposed to showing their independent performance on gene expression. For example, I would like to see scatterplots that show the competition ratio on one axis and other metrics (including mean) on the other, similar to ED Figure 16. Conceivably, one could add the third dimension of gene expression (or at least expression quantile) using color.

Response: Thanks for the suggestions. As suggested by the reviewer, we made the following 'colored' scatter plots, in which methylation concurrence ratios are on the X-axis, other methylation metrics are on the Y-axis, and gene expression quantiles are indicated by different colors (**Review Figure 3**). To clearly show the trend, we divided the genes into ten subgroups based on their values of methylation metrics, and the proportions of every expression quantile in

each subgroup are at the sides of the scatter plots. As shown below, methylation concurrence presents the best linear relationships with gene expression, as the proportions of higher expressed genes are gradually decreased when the concurrence ratios increase.

Review Figure 3. The scatter plots compare methylation concurrence and other methylation metrics in terms of gene expression correlation.

For each panel, the methylation concurrence ratios are on the X-axis. Other methylation metrics (i.e., average methylation, Entropy, Epipolymorphism, PDR) are on the Y-axis. Colors indicate the gene expression quantiles.

4. Deeper understanding of how competition distinguishes or identifies differentially regulated regions in cancer: The authors show that the competition ratio can identify dynamically regulated regions between normal and cancer samples in two matched tissue/tumor biopsy samples. However, the difference between these regions and those that can be identified by mean methylation is limited to gene set enrichment and overlap with a limited set of predefined tumor suppressor genes. In my opinion, the relevance of these competition-only targets is still largely speculative. Based upon visual inspection of the P1, P2, and P3 subsets, it seems like mean methylation is still quite highly associated with gene repression (as indicated by the tumor boxplots going to near zero). In contrast, the

competition ratio-only subset (P1) appears to be more false positive prone (the overall expression of these genes in tumor remains higher than for P2 and P3). Similarly, the change in the competition ratio appears to be more minimal in P1 than in P3. It very much appears that competition may be more sensitive to subtle changes in DNMT activity, but therefor requires a more careful statistical delineation between significant and non-significant.

Response: Thanks for the suggestions. Because there is no replicate for WGBS data in the matched normal/tumor biopsy samples for UCEC and BRCA, the classical p-value-based hypothesis testing framework for identifying differential signals does not apply. To identify the statistically reliable difference, we use the local false discovery rate (local-fdr), a statistical criterion that assesses the credibility of individual discoveries under the Bayesian framework (Efron, 2004), to solve this problem.

Let d_j denote the difference between matched tumor and normal samples based on any methylation measure (e.g., average methylation) for gene j , $j = 1, \dots, p$. Local-fdr assumes that genes come from two populations: differential and non-differential. Let p_0 denote the prior probability that a gene is non-differential. Let $f_0(d) := P(D = d | \text{non-differential})$ and $f_1(d) := P(D = d | \text{differential})$ denote the conditional probability density of D at d given that D comes from the non-differential and the differential gene population, respectively. Thus, by Bayes' theorem, the posterior probability of a gene being non-differential given its summary statistics is $P(\text{non-differential} | D = d) = p_0 f_0(d) / f(d)$, where $f(d) := p_0 f_0(d) + (1 - p_0) f_1(d)$ is the marginal probability density of D . The local-fdr of gene j is then defined as $local-fdr_j = \frac{f_0(d_j)}{f(d_j)}$.

Because $p_0 \leq 1$, $local-fdr_j$ is an upper bound of $P(\text{non-differential} | D = d_j)$. In the methylation field, the conventional threshold for methylation difference is 10% (Farlik et al., 2016). To control this posterior probability, we simply control the local-fdr under the same value. Notably, local-fdr differs from the traditional FDR in that FDR ensures the reliability of the identified differential genes as a set while local-fdr focuses on the reliability of each identified differential gene. To implement local-fdr, f_0 is assumed to be normal distributed and is estimated from middle-ranged d_j 's and f is estimated nonparametrically from $\{d_1, \dots, d_p\}$, as in the R package 'locfdr'.

Using the local-fdr method, between UCEC normal and tumor, we identify 2,557 genes, 2,056 genes, and 871 genes whose promoters take significant changes of average methylation, methylation concurrence, and methylation variation, respectively. Because the threshold defined by local-fdr is more stringent than the conventional cutoff (10% difference) we used before, we identified much fewer differentially methylated genes, and the new P1/P2 genes present more substantial expression differences (**New Figure 3c**). We notice that, although the P2/P3 gene expression is lower than P1 in the tumor sample, the repression of P1 genes is more substantial. The medians of P1 genes decrease more than 2 folds ($\log_2\text{Fold} = -1.11$), while the decrease of P2/P3 genes are less notable (P2, $\log_2\text{Fold} = -0.85$; P3, $\log_2\text{Fold} = -0.63$). And the observation in BRCA normal/tumor pair is also the same (**New Figure S12c**).

New Figure 3c. Gene expression change of P1, P2, P3, and 1,000 randomly selected genes in UCEC normal and tumor samples. The fold changes between median values (log2 scale) are indicated below. The two-tailed Wilcoxon signed-rank test was used for the significance test.

New Figure S12c. Gene expression change of P1, P2, P3, and 1,000 randomly selected genes in BRCA normal and tumor samples. The fold changes between median values (log2 scale) are indicated below. The two-tailed Wilcoxon signed-rank test was used for the significance test.

Also, given the prior claim that the competition score shows a greater correlation with gene expression within a sample (as discussed above), I don't see a discrete claim here about the precision of this score to predict changes in gene expression instead of being merely associated. Again, the boxplots in Figure 3c and ED Figure 8c show general trends that support some overlap between differential gene expression, competition and/or mean promoter methylation. However, what is the overlap between repressed genes and promoter sequences that are differential according to these two scores? What is the relationship between change in competition and change in expression and how does this differ from mean? One could imagine showing this with scatter plots showing the difference in promoter methylation and gene expression between cancer and normal.

Response: Thanks for the suggestions. To show the correlation between reliable gene expression change and reliable methylation difference between cancer and normal, we utilized

paired normal and cancerous lung samples, which have three replicates for both methylation and gene expression data (GSE70091)(Li et al., 2016). As indicated in **New Fig S8**, the differences (Δ) of methylation concurrence are also negatively correlated with gene expression change (left panel) and outperform the differences of average methylation (middle panel) and methylation variation (right panel).

Supplementary Figure 8. The correlation between gene expression change and promoter methylation difference. Scatter plots showing the correlation between gene expression change and the differences (Δ) of methylation concurrence (left panel), average methylation (middle panel), and methylation variation (right panel) between lung cancer and normal lung samples. All genes' promoters are included for analysis, but only those exhibiting a significant methylation change between cancer and normal are plotted. This results in 3,759 promoters for methylation concurrence, 3,956 promoters for average methylation, and 575 promoters for methylation variation. Pearson's correlation was used. P-values were calculated by the two-tailed correlation test. LOWESS lines were plotted to describe the relationships between variables (indicated by red curves).

5. Description and analysis of high and low competition canyons. To me, the current description and findings associated with the competition ratio at Polycomb regulated (p) versus active (a) canyons is largely not novel and distorted by a few missing variables/descriptions. For example, large domains of unmethylated DNA are apparent for Polycomb regulated targets, whereas smaller canyons are associated with CpG islands at highly transcribed genes. This is born out by the author's analysis of public CD+ T cell data: aCanyons are highly enriched for euchromatic chromatin modifications associated with transcription (H3K4me3) and are highly transcribed. It's therefor not surprising that they would show low "competition" (discordance) within the gene body, as the gene bodies of actively transcribed genes are in general very highly methylated. The negative correlation between gene expression and competition in Figure 4d is very likely explained by these gene bodies being very highly methylated, and cannot be captured by the competition score. At the very least, the authors should include data showing the mean methylation tracks for aCanyon and pCanyon genes to give the reader a frame of reference.

Response: Thanks for the suggestions. To figure out whether the negative correlation between aCanyon target gene expression and methylation is due to their highly methylated gene-body, we added the average methylation tracks of Canyon genes, as well as two randomly selected genes as control groups, i.e., randomly selected highly expressed genes (purple in **New Figure 4**) and lowly expressed genes (yellow in **New Figure 4**). As shown below, although 'aCanyon targets' and 'random highly expressed genes' are both highly expressed and highly methylated at gene-body, their functions and correlation patterns are distinct. These data indicate the aCanyon gene group segregated by methylation concurrence has unique regulation models.

New Figure 4. (b) Expression of aCanyon target genes (red), pCanyon target genes (blue), 1,000 randomly selected highly expressed genes (purple), and 1,000 randomly selected lowly expressed genes (yellow). (c) Functional enrichment analysis of the four gene groups in (b). Enriched gene counts in each group are indicated on the left side. P-values were measured by two-tailed Fisher's Exact Test and adjusted by the Benjamini-Hochberg method. (d) Spearman correlation between gene expression and average methylation of 100bp-bin in gene regions. Totally 120 bins from -2kb to 10kb were measured. (e) The profiles of average methylation for each gene group. The standard deviations are indicated by the width of the shaded area.

I suspect that pCanyons are going to be substantially larger than aCanyons, consistent with an extended Polycomb footprint and the fact that valley prediction will likely truncate abruptly at the boundary of a transcribed gene body (similarly, the position of genes relative to the valley boundary may be more obvious). These additional genomic features can also be represented using composite style plots such as in Figure 4a.

Response: Thanks for the suggestions. The pCanyons are indeed longer than aCanyons (**New Figure S16b**).

New Figure S16b. The length of aCanyons and pCanyons in stem cells (H1) and CD3+ T-cell.

Normalized figures such as those associated with Figure 4a would be misleading if the length of sequences between “left” and “right” are structurally different.

Response: Thanks for the suggestions. For the composite plots in Figure 4a, instead of normalizing all Canyons to 5kb, now we present the uncompressed profiles around canyon borders. Please see the **New Figure 4a** on **Page 5** of this rebuttal letter.

Similarly, differences in length would bias the results described in ED Figure 13, as pCanyons will have more bases from which to capture an interaction. If this is the case, the “random” background would also have to be adjusted for the unique features of each subgroup.

Response: Thanks for the suggestions. As indicated in **New Figure S16b**, the pCanyons are longer than aCanyons. To exclude such bias for the chromatin interaction analysis, we added two control groups of random regions whose lengths are equal to aCanyons and pCanyons, respectively. As indicated in **New Figure S16d**, the longer control group is more enriched of interactions. But the enrichment in pCanyons is still higher than both control groups.

New Figure S15d. The chromatin interactions on methylation canyons. Percentages of canyons and random genomic regions that have high-order chromatin interactions in CD3+ T-cell. Chromatin interactions were defined by pairs of Hi-C anchors. ‘Self-interacting’ indicates the anchor pairs located in the same canyon/random region. ‘Distant-interacting’ means only one of them is in the canyon/random region.

This issue may also hold for the analysis of TF binding preferences. If the background of pCanyons and aCanyons is structurally different (ie. if aCanyons are more focally restricted to the TSS of actively transcribed genes whereas pCanyons are larger and less well positioned with respect to the TSS of the relevant gene), you may expect differential enrichment of generic transcriptional activators, such as TAF1 and TBP, purely as a bioinformatic artifact. I would be curious if these TF enrichments hold if motif analysis is fixed to the TSS (or CpG islands) of genes that reside within Canyons, as opposed to scanned agnostically across the entire domains (which I believe from the Methods is what’s happening).

Response: In the original Figure 4e and Figure S12, the analysis of TF preference was focused on Canyon regions. Following the reviewer’s suggestions, we added the TF motif enrichment analysis at promoters of Canyon genes. And the definition of promoter in this study is the regions from 1kb upstream to 500bp downstream of TSS. So, the new analysis is fixed to TSS proximal regions and could avoid the length and structural difference between aCanyons and pCanyons. Similar to the original Figure 4e, the fold enrichment of TF motifs at promoters is also

negatively correlated with their mCpG preference, which agrees with our previous observation (**New Figure S16e**).

New Figure S16e. Relationship between fold enrichment at Canyon gene promoters and 5mC preference of TF motifs. Each dot represents a motif. Y-axis indicates the fold change (log2) between enrichment at aCanyon gene promoters and enrichment at pCanyon gene promoters of the same motif. The X-axis shows the 5mC preference of motifs measured by the SELEX technique. ‘methyl-plus’ TFs prefer to bind methylated sequences, while binding of ‘methyl-minus’ TFs are not favored by 5mC. Spearman’s rank correlation was used. P-values were calculated by the two-tailed correlation test for Spearman’s correlation. The linear model was plotted to describe the relationships between variables (indicated by the dashed line).

We also analyzed the TF binding preference at Canyon gene promoters using ChIP-seq data in human stem cells (**New Figure S16c**). Again, similar to the original Figure S12, aCanyon gene promoters are more enriched with transcription activators, such as TAF1 and TBP. In contrast, pCanyon gene promoters are more enriched with the Polycomb complex, such as EZH2 and SUZ12.

New Figure S16c. The transcription factor binding difference at canyon gene promoters Scatter plot showing TF binding difference between two groups of Canyon gene promoters in H1 human stem cells. The X-axis indicates the odds ratio between TF-bound-aCanyon gene promoters (%) and TF-bound-pCanyon gene promoters (%). For the left part (blue shaded), the Y-axis is the percentage of pCanyon gene promoters bound by TF. For the right part (red

shaded), the Y-axis is the percentage of aCanyon gene promoters bound by TF. Each dot represents a TF.

Minor comments:

1. The statement in the introduction: “For example, the DNMT3A and TET2 double-knockout mice show worse survival than single-knockout counterparts,” should qualify that this is in conditional knockout specific to the hematopoietic system.

Response: Thanks for the suggestions. We have changed this statement as below.
“For example, in a conditional knockout study in mouse hematopoietic system, the Dnmt3a and Tet2 double-knockout mice show worse survival than single-knockout counterparts”.

2. I am curious how the TET and DNMT3A KOs impact other read-level methylation estimates beyond mean and the competition ratio. As I understand it, the competition ratio is similar to other metrics such as PDR, Epipolymorphism, or Entropy, in that it should be higher when read-level methylation is maximally entropic and lower as it approaches fully unmethylated or methylated states. One may expect similar trends for these metrics: while the mean methylation direction changes in agreement with the enzyme class deleted, these other metrics may show trends that resemble the competition ratio.

Response: As suggested by the reviewer, we analyzed the methylation variation scores (Entropy, Epipolymorphism, and PDR) in *Dnmt3* and *Tet* knockout mouse ESC. As indicated in **Review Figure 4**, methylation variation is substantially decreased in the *Dnmt3* KO sample, which is similar to average methylation and methylation concurrence. But in the *Tet* KO sample, they behave differently. PDR shows a similar decreasing trend as methylation concurrence. This is expected as PDR is the ratio of partially methylated reads and may carry some methylation concurrence information. But Entropy and Epipolymorphism are not. They both increase in *Tet* KO and behave similarly to average methylation.

Review Figure 4. The methylation variation (Entropy, Epipolymorphism, or PDR) altered regions in *Dnmt3* and *Tet* knockout mouse ESC.

3. The schematic in Figure 1a does an adequate job of describing the nature of data that informs DNMT3 vs. TET competition, but does not relate how these data are used to calculate the score itself. More detail should be provided to show how simulated data such as this relates to the unweighted and weighted competition scores. For inspiration, the authors could consider Figure 2 from Sherer et. al Nucleic Acids Research 2020 (Ref. 57) or Figure 2 in Guo et. al Nature Genetics 2017 (Ref. 14).

Response: Thanks for the suggestion. We modify the **New Figure 1a** as below. It elucidates how the methylation concurrence is calculated. To show how the weighted and unweighted versions of methylation concurrence are calculated and how they are different from the other

metrics, we add **New Figure S1**, which compares the different methylation quantifications in various configurations.

New Figure 1a. Schematic of DNA methylation concurrence captured by bisulfite-seq. Solid circles are methylated cytosines. Blank circles are unmethylated cytosines. Red circles are unmethylated cytosines in partially methylated reads, i.e., methylation-concurrence cytosines. The equation below shows the calculation of methylation concurrence ratio using the example above.

New Figure S1. The definition of methylation concurrence ratio, and its comparison with different methylation metrics. (a) The calculations of the weighted and unweighted versions of methylation concurrence. Solid circles are methylated cytosines. Blank circles are unmethylated cytosines. Red circles are unmethylated cytosines in partially methylated reads, i.e., methylation-concurrence cytosines. Equations below show the calculation of weighted and unweighted versions of methylation concurrence ratio using the example above. ‘frags’ is short for ‘fragments’. (b) The calculations of different methylation metrics using examples of bisulfite-seq reads, which cover a 4-CpG region.

4. What do the error bars represent in plots Figure 2b, and ED Figures 1, 2, 3, 4, and 6?

Response: We want to apologize for this misleading. The bars in the original ‘binned/grouped’ plots are not error bars. They indicate the upper quantiles and lower quantiles of each group. And the dots between them indicate the median of each group. To avoid such misleading and address the reviewer’s 3rd major concern, we decided to replace all the “binning/grouping” style plots with the scatter plots, which show the unbinned raw data points.

5. Could the authors provide a more rigorous explanation regarding why the competition ratio is structurally different than other variation metrics for distinguishing aCanyons and

pCanyons? Qualitatively, I agree that it doesn't preserve transition peaks around the boundaries of pCanyons, but really these other metrics also indicate that canyon boundaries don't differ between classes, so I'm not clear on the added value (if you plotted the difference between aCanyons and pCanyons, I'm not sure competition would look much different).

Response: Thanks for the suggestions. To check whether methylation concurrence is a better classifier of aCanyons and pCanyons, we plotted the difference (Δ) between pCanyons and aCanyons for the different methylation metrics. As indicated in **New Figure S15**, the difference in methylation concurrence is much more substantial than those in methylation variation scores. The difference inside Canyon is ~ 2.8 -fold higher than the flanking regions. But for methylation variation, this fold decreases to 1.5 for Entropy, 1.8 for Epipolymorphism, and 1.3 for PDR (**New Figure S15b**). These data indicate methylation concurrence can better distinguish aCanyons and pCanyons.

New Figure S15. The profiles of methylation concurrence and methylation variation scores at two canyon groups. (a) Average profiles of methylation concurrence, methylation Entropy, Epipolymorphism, and PDR on methylation canyons in CD3+ T-cell. 'aCanyons' (red) are low-concurrence canyons. 'pCanyons' (blue) are high-concurrence canyons. The X-axis indicates the distance to canyon borders. (b) The difference (Δ) between pCanyons and aCanyons for scores in (a).

Response to Reviewer #2:

Reviewer #2: Shi and colleagues describe a new DNA methylation metric termed 'methylation competition' that appears to be better correlated with gene expression, silencing of tumor suppressor genes and segregates DNA methylation canyons into active and

polycomb repressed canyons (aCanyons and pCanyons). Overall the manuscript has several interesting aspects but can be further improved in clarity and presentation.

Response: We thank the reviewer for acknowledging the significance and novelty of our findings.

Specific comments:

There is not enough evidence provided that methylation competition is involved in tumorigenesis as the title might suggest. The metric might explain some gene expression observations better but that is something else.

Response: Thanks for the suggestions. We have changed the title to “*The Concurrence of DNA Methylation and Demethylation is Associated with Transcription Regulation*”.

Similar in the abstract, the statement ‘during tumorigenesis’ and ‘important for... tumorigenesis’ would require more data than comparing the presented cancer and normal data.

‘Largely distinct’ is a rather vague ending for the abstract.

Response: Thanks for the suggestions. We have removed these vague statements and re-edited these sentences in the abstract as below.

“In particular, the elevation of methylation concurrence is associated with the repression of 40~60% of tumor suppressor genes, which cannot be explained by promoter hypermethylation alone. ... Together, methylation concurrence represents a novel methylation metric important for transcription regulation and is distinct from conventional metrics (e.g., average methylation and methylation variation).”

The metric ‘methylation competition’ is central to the paper but not well explained or introduced. Figure 1a explains the hypothetical bisulfites sequencing reads but should be expanded and refined to actually introduce the metric.

Response: Thanks for the suggestions. To elucidate how the methylation concurrence is calculated and how it is different from the other metrics, we add **New Figure 1a** and **New Figure S1**. Please check them on **Page 21** of this rebuttal letter.

The text mentions (line 91) consecutive C, but that is not well explained.

Response: Thanks for the suggestions. We have re-edited these sentences as below.

‘Hence, we dissect bisulfite sequencing reads into three categories of fragments (or sub-reads), i.e., methylated fragments which consist of consecutive methylated CpG(s) (solid circles in Fig 1a, denoted as ‘M’), unmethylated fragments which are the fully unmethylated reads (blank circles, denoted as ‘U’), and methylation concurrence fragments which are segments of unmethylated CpG(s) in partially methylated reads (red circles, denoted as ‘C’). We define the ‘methylation concurrence ratio’ of a genomic region as the sum of methylation concurrence fragments’ weights divided by the sum of all fragments’ weights in that region.’

It also ignores the competition that occurs on the filled circles that are the result of 5hmC (which will be called as mC in the data). Especially in the TET targeted regions one would expect this to be not insignificant.

It also doesn't account or consider other means of creating the partial methylation on fragments, such as Dnmt1's failure to maintain all methylation post replication or other factors that could interfere with Dnmt1 activity.

Response: Thanks for the suggestions. We add a new paragraph in the Discussion to clarify the limitations of the current study.

'There are limitations for the methylation concurrence metric. First, 5mC can be oxidized to 5-Hydroxymethylcytosine (5hmC) by the TET proteins, but the conventional bisulfite-seq cannot distinguish 5hmC from 5mC, so the methylation concurrence ratio is an underestimation of the real concurrence. This can be improved by applying the oxidative bisulfite-seq, which can discriminate between 5mC and 5hmC(Booth et al., 2012). Second, the current study only associates methylation concurrence ratio with the co-localization of de novo methyltransferases DNMT3A/B and active demethylase TET. The maintenance methylation by DNMT1 and passive demethylation due to DNA replication are not discussed. In future work, a more comprehensive model is needed to incorporate all these factors.'

The difference to the other variation metrics could be explained better. As the authors state they are distinct in some aspects, but other behaviors are rather similar. This is related to the above point in the abstract.

Response: Thanks for the suggestions. We would like to reiterate the uniqueness of methylation concurrence compared to the variation metrics. First, methylation concurrence is the only metric that correlates with DNMT&TET co-localization (**Figure 1b, 1c, 1d, Figure S2, and Figure S4**). Second, it is strongly associated with gene expression and outperforms variation metrics (**Figure 2 and Figure S7**). And such association is consistent for different regulatory elements, e.g., promoter, gene-body, and enhancer. Third, the elevation of methylation concurrence but not variation could be used to explain the repression of 40~60% of tumor suppressors, which cannot be unveiled by promoter hypermethylation (**Figure 3 and Figure S12**).

The authors use mESCs that express also high levels of Dnmt3b and other TETs, however it is never explained why this is just focused on the Dnmt3a and TET1 competition.

Response: Thanks for the suggestions. We believe that methylation concurrence can also be used to quantify the concurrence effect of other DNMT3 and TET enzymes. To validate this hypothesis, we analyzed the ChIP-seq data of DNMT3B and TET2 in mouse embryonic stem cells. Similar to previous results for DNMT3A and TET1, the binding intensity of both DNMT3B and TET2 are positively correlated with methylation concurrence (**new Figure S4a, S4b**). Then we check the co-localization of different DNMT&TET combinations. As shown in **new Figure S4d, S4e, S4f**, similar to the 'DNMT3A&TET1', the co-localization of 'DNMT3A&TET2', 'DNMT3B&TET1', and 'DNMT3B&TET2' are all positively correlated with methylation concurrence. These data suggest that methylation concurrence can delineate the concurrence effect of other DNMT3 and TET enzymes.

New Figure S4. The methylation concurrence can delineate the concurrence effect of other DNMT3 and TET enzymes in mESCs. The binding intensities of both (a) DNMT3B and (b) TET2 are positively correlated with methylation concurrence at gene promoters. The (c) 'DNMT3A&TET2', (d) 'DNMT3B&TET1', and (e) 'DNMT3B&TET2' joint-regulation scores are positively correlated with methylation concurrence.

Line 143 the knockouts are described but this is very confusing as it states Tet knockout and ONLY the legend clarifies that this a Tet1, 2, 3 triple knockout.

Response: Thanks for the suggestions. We have added a detailed description of knockout samples into the figures and figure legends.

ZBTB16 is used as an example for a TSG, but no details (methylation data) for its promoter are provided. Its not clear which data support the strong conclusion 'that methylation competition at its promoter indicate a novel mechanism for its tumor suppressing role.'

Response: Thanks for the suggestions. We add the methylation data of the *ZBTB16* gene in **New Figure S11a**, as well as the read level data.

New Figure S11a. The methylation concurrence, average methylation, and RNA-seq reads densities around *ZBTB16* gene locus in UCEC normal and tumor samples. The bisulfite-seq reads at the *ZBTB16* gene promoter region (yellow shaded) are shown below.

We agree that we don't have solid evidence for the tumor suppressor role of *ZBTB16* in uterus tumor. So, we have removed the strong conclusion and re-edit these sentences as below.

'Although ZBTB16 has an unclear function for uterine cancer, a previous study reveals that overexpression of ZBTB16 inhibited proliferation in cervical carcinoma cells and induced apoptosis (Rho et al., 2007). Given the strong association between high methylation concurrence in promoter regions and reduced gene expression, these results suggest that ZBTB16 may also act as a tumor suppressor in uterine tumors as well, although additional functional studies are needed to test this hypothesis.'

Line 274, H3K27me3 is a histone modification, so bound by H3K27me3 is incorrect and should say 'enriched'. Also line 372.

Response: Thanks for the suggestions. We have changed the words in the following sentences.

*'... we refer to the high-concurrence canyons, which are **enriched with** H3K27me3, as Polycomb canyons (pCanyons).'*

*'A previous study of mouse hematopoietic stem cells (HSCs) finds that some canyons are active while the others are silent, and that such a difference is explainable by the **enrichment** of H3K4me3 or H3K27me3.'*

1f,g: Add CGIs to the tracks

Response: Thanks for the suggestions. We have added the CpG island (CGI) track into **New Figure 1f and 1g**, as well as in **New Figure S3**, which shows the read level data. And please note that there is no CGI nearby gene *Ogt*.

New Figure 1. (f) UCSC Genome Browser tracks show DNMT3A1 binding (orange), TET1 binding (purple), methylation concurrence (red), average methylation (black), methylation variation (blue), and gene expression data (green) at *Prok2* gene. (g) Same as (f), but for gene *Ogt*. CpG islands are shown in grey.

1f: It would be helpful to see the read level data for a regions like this.

Response: Thanks for the suggestions. We have added the bisulfite-seq reads level data at the promoter-proximal regions of gene *Prok2* (**New Figure S3a**) and *Ogt* (**New Figure S3b**).

1g: The average methylation shows a smaller peak in the unmethylated canyon, that should lead to some higher variation but doesn't show up in the track below.

Response: As indicated in **New Figure S3b**, there is a methylated region inside the unmethylated canyon (yellow shaded). But none of the reads in this region covers more than 4 CpGs. Due to the limitation of the methylation variation algorithms (e.g., Entropy), these reads are not utilized, and thus the variation score is not able to be calculated.

New Figure S3. (a) The bisulfite-seq reads at *Prok2* gene promoter. (b) Same as (a), but for gene *Ogt*. Each consecutive horizontal sequence of circles represents a bisulfite-seq read. Solid circles are methylated cytosines. Blank circles are unmethylated cytosines. Red shaded circles are unmethylated cytosines in partially methylated reads, i.e., methylation-concurrence cytosines. CpG islands are shown in grey.

1h: Dnmt3a KO in ESC should retain most methylation given high levels of Dnmt3b. More details should be provided on the knockouts and KO data.

Response: Thanks for the suggestions. According to the original paper of this data, they reintroduced DNMT3B1 into stem cells that lack DNA methylation due to deletions of all *Dnmt* genes (*Dnmt3a1*, *Dnmt3a2*, and *Dnmt3b1*). They also found that genomic binding of the reintroduced DNMT3B1 in knockout cells resembles that in wild-type ES cells (Baubec et al., 2015). We have added these details into the legend of Figure 1.

1i: TET ko are quite similar in depleted and elevated regions. These should be triple knockout (see comment above about the confusing nomenclature that needs to be fixed in text and the Figure panel) and one should expect all the C cytosines to turn into M if one follows the authors logic, so it is surprising to see so many competition elevated regions.

Response: We would like to explain that not all the unmethylated cytosines can be turned to methylated cytosines in *Tet1*, *Tet2*, *Tet3* triple knockout (*Tet* TKO). In some regions, the methylation concurrence can be elevated when the average methylation increased. As an example, in the **Review Figure 5** below, with the increased average methylation (from 0.155 in WT to 0.345 in *Tet* TKO), some unmethylated parts become partially methylated, so the methylation concurrence is elevated from 0.131 to 0.251.

chr4: 88,920,000 – 88,930,000

mES WT: Average Methylation = 0.155 , Methylation Concurrence = 0.131

mES Tet TKO: Average Methylation = 0.345 , Methylation Concurrence = 0.251

- Methylated CpGs (M)
- Unmethylated CpGs (U)
- Methylation Concurrence CpGs (C, unmethylated CpGs in partially methylated reads)

Review Figure 5. An example shows that the methylation concurrence is elevated at some regions in *Tet* TKO mouse ESC.

Most other figures have few panels and lots of important points are in the supplement, this could be arranged more effectively. In turn, panels such as 2b don't have to be in the main Figures.

Response: Thanks for the suggestions. We have merged the panels into the same figure if they are related to the same topic. As Figure 2b highlights the consistent performance of methylation concurrence among different regulatory elements, we would like to keep it as a major observation.

Response to Reviewer #3:

Reviewer #3: The authors introduce a novel concept of methylation competition in an elegant way to identify regions that are bound by both methylation & demethylation agents within the same cell using bisulfite sequencing data.

Response: We thank the reviewer for acknowledging the significance and novelty of our findings.

The analysis on correlation with gene expression could benefit from more detail, how the results in Figure 2 are reported. Is this from all cells, from how many donors, etc. and across how many CpG sites/genes?

Response: Thanks for the suggestions. We add the following details to the legend of Figure 2.

'The WGBS data is from the Roadmap project with GEO accession number GSM1186660. The CD3+ primary cells are from a 37-year-old male. To increase reliability, we select regulatory elements whose CpGs are all sufficiently covered (≥ 4 reads). This results in 9,876 promoter regions, 14,868 gene-body regions, and 3,351 enhancer regions.'

Sequencing depth: It is not clear why the authors would expect the relationship to be correlated with sequencing depth? Isn't this an artefact?

Response: We would like to apologize for the misleading statement. We didn't expect the relationship to be correlated with sequencing depth. This sentence is changed as below.

"Using a down-sampling analysis, we observe that all of the three measures (methylation concurrence, average methylation, and methylation variation) are negatively associated with gene expression at sufficient sequencing depth."

Tumor suppressor genes:

What is the interpretation that the three groups of genes in Figure 3a (P1, P2, P3) are enriched in different pathways? Any thoughts on biological interpretation of this finding?

Response: Thanks for the suggestions. As Reviewer #2 mentioned, we don't have enough evidence to prove the role of methylation concurrence in tumorigenesis directly. Therefore, we remove "tumorigenesis" from the title, and then we found this part of function analysis is purely speculation and does not relate to the topic of this paper. To avoid misleading, we have removed this part in the revised version.

Finally, can do authors provide more interpretation what the methylation competition metric's biological interpretation is? Any thoughts on what this all means?

Response: We want to reiterate that the methylation concurrence metric delineates a new type of local epigenetic dynamics, which is associated with the simultaneous regulating of methyltransferases and demethylases. It is the only metric that correlates with DNMT&TET co-localization (**Figure 1** and **Figure S4**). Second, it is strongly associated with gene expression and outperforms average methylation and methylation variation (**Figure 2, Figure S7, and Figure S8**). Finally, it can disclose more methylation abnormalities and cancer drivers (**Figure 3** and **Figure S12**).

References

Baubec, T., Colombo, D.F., Wirbelauer, C., Schmidt, J., Burger, L., Krebs, A.R., Akalin, A., and Schübeler, D. (2015). Genomic profiling of DNA methyltransferases reveals a role for DNMT3B in genic methylation. *Nature* 520, 243–247.

Booth, M.J., Branco, M.R., Ficz, G., Oxley, D., Krueger, F., Reik, W., and Balasubramanian, S. (2012). Quantitative sequencing of 5-methylcytosine and 5-hydroxymethylcytosine at single-base resolution. *Sci New York N Y* 336, 934–937.

Charlton, J., Jung, E.J., Mattei, A.L., Bailly, N., Liao, J., Martin, E.J., Giesselmann, P., Brändl, B., Stamenova, E.K., Müller, F.-J., et al. (2020). TETs compete with DNMT3 activity in pluripotent cells at thousands of methylated somatic enhancers. *Nat Genet* 1–9.

Efron, B. (2004). Large-Scale Simultaneous Hypothesis Testing: The Choice of a Null Hypothesis. *J Am Stat Assoc* 99, 96–104.

Farlik, M., Halbritter, F., Müller, F., Choudry, F.A., Ebert, P., Klughammer, J., Farrow, S., Santoro, A., Ciaurro, V., Mathur, A., et al. (2016). DNA Methylation Dynamics of Human Hematopoietic Stem Cell Differentiation. *Cell Stem Cell* 19, 808–822.

Li, X., Liu, Y., Salz, T., Hansen, K.D., and Feinberg, A. (2016). Whole-genome analysis of the methylome and hydroxymethylome in normal and malignant lung and liver. *Genome Res* 26, 1730–1741.

Rho, S.B., Chung, B.M., and Lee, J. (2007). TIMP-1 regulates cell proliferation by interacting with the ninth zinc finger domain of PLZF. *J Cell Biochem* 101, 57–67.

REVIEWER COMMENTS

Reviewer #1 (Remarks to the Author):

I would like to thank the authors for all of their effort to add texture to this manuscript. I agree that “concurrency” is a more appropriate term for the metric in question and feel the additional figures, text, and analyses add improved resolution and depth that will benefit reception by the field. All told, I find the association between this new metric with gene expression, differential regulation across tumorigenesis, and the epigenetic regulation of unique local territories (canyons) to be intriguing, and there are certain benefits to this methodology compared to other read-level methylation-based conventions. However, I still feel the language of this manuscript overstates the conceptual advance of this method compared to others (often stating what it finds that other metrics do not and attributing it to mechanism), overinterprets the connection between read-level heterogeneity and the dynamic interplay between DNMTs and TETs specifically, and often assigns new functions from largely correlative observations. In most cases, my recommendation is that the method and findings are worth reporting, but that the language requires additional tuning. I believe this should heighten the impact of the manuscript by highlighting the value of this approach for finding and evaluating epigenetically dynamic regions, which I believe is its strongest selling point.

I am still not convinced about the conceptual uniqueness of methylation concurrency compared to other read-level methylation metrics such as entropy or PDR (the proportion of discordant reads), which also prioritize local, read-level heterogeneity in methylation state as a read out for dynamic regulation. Instead, it seems more to me like concurrency has a more carefully bounded dynamic range, and some of this may differ only because the authors apply conventions to prior metrics out of expedience that they do not for their own.

For example, in Supplementary Figure 2, it appears to me that the positive relationship between entropy/epipolymorphisms and expression holds for the majority of promoter regions, but these are “squished” to the far left of the plot (the LOWESS regression also suggests a positive correlation within this portion of the plots, where the majority of the data is located). These trends (and the correlations) appear born out by the minority of outlier promoters that dominate the dynamic range of these estimates. To me, concurrency does not look so different, other than that these outliers are more constricted. The relationship between concurrency and entropy to gene expression is also highly similar in Figure 2b and Supplementary Figure 7, and the overall enrichment change in the boundaries of canyons is also similar in Supplementary Figure 15. If I am reading the methods correctly, the calculations for entropy, epipolymorphism and PDR all pre-filter on reads where at least 4 CpG strings can be measured in phase. As the authors show in their new track figures, that would suggest that their new metric benefits by considering and including reads in a different manner than for these other approaches. However, this is only a difference in convention and not in concept (this is also stated by the authors in their comments to Reviewer 2 regarding the OGT locus). Getting 4 CpGs in the same sequencing read is also far less likely outside of CpG islands, which would mean that more global claims are displayed as a fair comparison, but actually “mask” or remove substantial data for these other metrics. This is basically accommodating somewhat arbitrary conventions for how they are typically used and are not fair comparisons.

Entropy, PDR, and epipolymorphisms are conceptually quite similar to concurrency. At the very least, I think the language of the text should more carefully establish the fact that, as a metric, concurrency is designed to maximize detection of heterogeneous methylation according to a similar concept as these other approaches, as opposed to suggesting that these other methods don't cover very similar intellectual territory. This is particularly important because the language currently suggests that the authors treat each metric fairly, as opposed to applying empirical conventions to historical metrics that compromise the comparison to their new one.

Additional Comments:

Within the abstract, the second sentence is ambiguous and overestimates the audiences understanding of the term “concurrency.” Possibly this could be changed to something like “Although the dynamics of either methylation or demethylation have been intensively studied in

the past decade, the direct effects of their interaction on gene expression remains elusive.”

As to the major point above, Supplementary Figure 1b should clarify the meaning of N/A to indicate that Shannon entropy and epipolymorphisms are not calculated for this right-most class of reads (though they could be if a different convention were applied). Similarly, would PDR as calculated in this manuscript consider these 3 and 2 CpG strings (it does not appear to be the case from the methods)? These differences in convention may also impact the ability for metrics like Entropy to be fairly applied to LCP-like loci as shown in Supplementary Figure 9 (≥ 4 CpG containing reads will be more rarely captured in this regime).

The authors claim to have conducted an integrative analysis of 85 methylomes and 44 transcriptomes within the introduction, but the manuscript itself still favors single samples or limited data sets to make select points (a major comment of the first review that is not really addressed). The authors may have processed this many samples for their new metric, but this is not the same as an integrative analysis and misleading about what the authors have actually done. The language of this should be changed and possibly included moved to the discussion as a provided resource to the community.

I very much appreciate the change in language to highlight the intriguing nature of the authors' findings without oversuggesting a mechanistic connection. However, across their functional comparisons, the change in concurrence is much more convincing for DNMT3 KO than TET KO contexts, suggesting additional factors specifically impact the negative regulation of DNA methylation. The new text in the conclusions around these points should highlight this particular discrepancy. For example, “Notably, the change in concurrence only requires that DNA methylation be heterogeneous on the same molecule, and is more sensitive to changes in DNMT than TET activity, suggesting additional mechanisms may act to negatively regulate this modification.”

The association between changes in gene expression and DNA methylation as calculated by concurrence, average methylation, or entropy are not particularly convincing (Supplementary Figure 8, requested in review). The comparison is between lung cancer and tumor-adjacent normal tissue from TCGA and the statements justified by Pearson correlation across the full dynamic range. Even visual inspection suggests that the relationship between promoter DNA methylation and expression changes is not cumulatively predictive. As most promoters that change their methylation status appear to gain DNA methylation as part of transformation, there is a clearer enrichment for the average methylation changes in the bottom right quadrant (increasing methylation that corresponds to gene repression). This strikes me as the most convincing result of this test. However, the language of the text associated with this figure instead concludes this is supportive of concurrence as the most predictive DNA methylation metric for gene expression. I think that is a very cursory statement that overstates the conclusions drawn from this exploration.

The closing statement of the canyon section should be reworded. Although the authors “redefine” or “rediscover a- and pCanyons using the concurrence metric, I imagine they could be more easily and precisely identified using gene expression (or Polycomb enrichment). For example, “Together, concurrence shows that the dynamics of methylation and demethylation are unique for a- and pCanyons and strongly associate with additional distinguishing features of their regulation.”

All told, I find the added control features to the Canyon section a little unconvincing. The heightened negative Spearman correlation between gene body methylation and expression in aCanyons in Figure 4 could very well be the effect of the difference in the dynamic range for gene expression in this set compared to the “random highly expressed” background (Figure 4b). Canyons are pre-selected as large undermethylated regions, which did not utilize concurrence in their discovery (only their characterization). This difference between the Spearman correlations in aCanyons versus very highly expressed genes could effectively reflect bias between the aCanyon expression distribution and that of the “very highly expressed gene” sets.

Because the authors flip from defining canyons to showing behavior centered around TSSs, I would like a third set of plots below Figure 4d and 4e that shows where the canyon boundaries are in relationship to the TSS and gene body (eg. %fraction Canyon on the y axis).

The authors should also consider the strength of their conjecture about the relationship versus

correlation of a- and pCanyons with the affinity of different TFs for methylated DNA (which is derived from in vitro Selex experiments without additional chromatin contexts that likely dictate binding). The speculation of how the binding of these factors might impact changes in gene expression ignores the fact that many of the pCanyon enriched factors (Ezh2, Suz12, Rest) are associated with repressive complexes. Again, the fact that these canyons have differential enrichment of methyl-sensitive TFs is novel and can be reported, but the conclusions drawn from that association should mostly end with the idea that these classes appear to be differentially regulated even in the steady state.

Similarly, in Supplementary Figure 15, it's not clear to me how the random background was readjusted to make the claim that within-group interactions are higher in pCanyons than aCanyons, but the fact that the adjusting for length weakens the claim should be cause enough for caution. The authors should at least be clearer about how they select background features in their methods.

My comments that spurred Reviewer Figure 3 were related to the idea that there are going to be some contexts where concurrence will not be relevant for detecting or predicting transcriptional activity, but other metrics might be. For example, very highly methylated loci (promoters associated with many germline genes, certain retrotransposon classes such as IAPs that are highly DNA methylation dependent, or imprinted genes) will have a low concurrence precisely because they are very highly methylated and effectively maintained in a repressed state by DNMT1. The scatterplot of promoter DNA methylation and concurrence (top left-most plot in Reviewer Figure 3) highlights this point: a discrete class of promoters show low concurrence, are not expressed, and are very highly methylated. A very valuable next step for these authors or the field would be to work out ways to integrate different metrics that appear to capture distinct dynamics. It would be appropriate for this manuscript to both highlight this key point as well as note the potential for integrating different metrics as part of their Discussion.

Additional comments related to those made by Reviewer 2:

Reviewer 2 seems to share a similar skepticism that the differences between methylation concurrence and other read-level methylation metrics are not as dramatic as the authors claim, and in several instances, required additional "raw data" displays such as the new Supplementary Figure 3. These figures highlight how some of these discrepancies appear to be algorithmic and not conceptual, and are not always in alignment with author statements. For example, the reviewer highlights a peak in increased methylation concurrence within the Ogt locus that is not captured by variation. The authors address this concern by saying that the difference between variation and concurrence is in part explained by which reads are considered by the entropy algorithm, whereas concurrence considers many more reads. In this manner, many of the claims by the authors feel misleading for similar reasons as highlighted above. A reader would believe that the concept of entropy or read-level variation is not adept at capturing the same changes as concurrence, when in reality at least a part of the difference could simply be that the authors are including or excluding different data according to conventions (which are not applied equivalently to their own). Certainly these reads with fewer than 4 CpGs are exhibiting "variation" and show "increased entropy," and these terms have meanings beyond the algorithmic convention applied by the authors.

The authors should think more carefully about how to distinguish their metric and compare/contrast it to others more clearly. There is value in it, but there's no need to discount these other methods in their entirety.

Similarly, the Reviewer asks the authors to explain why the Tet KO cells do not show complete loss of concurrence. Notably, the authors use these KO data to experimentally justify their metric as representing conflicting methylation and demethylation within cells by DNMTs and TETs. Reviewer Figure 5 highlights that there are in fact very convincing instances where read-level heterogeneity increases in a manner that cannot be explained by the purported biochemical model posited by the authors. At the very least, the authors should account for this, include these figure panels, and explain them.

These and other instances highlight the need for more careful wording of the manuscript in general. In the track provided for the tumor suppressor ZBTB16, the authors note a strong association between high concurrence and transcriptional repression, though the values and nature of the trend are not particularly convincing (an increase from 0.124 to 0.232 for a metric whose dynamic range is only so clear, New Supplementary Figure 11a).

Reviewer #3 (Remarks to the Author):

All comments have been addressed.

Summary

First, we appreciate the perceived excitement among the reviewers as they clearly acknowledged the **significance and novelty** of our findings. For example:

- “I would like to thank the authors for all of their effort to add texture to this manuscript. I agree that “concurrency” is a more appropriate term for the metric in question and feel **the additional figures, text, and analyses add improved resolution and depth that will benefit reception by the field.**” (Reviewer #1)
- “I find the association between this new metric with gene expression, differential regulation across tumorigenesis, and the epigenetic regulation of unique local territories (canyons) to be **intriguing**, and there are **certain benefits to this methodology** compared to other read-level methylation-based conventions.” (Reviewer #1)
- “All comments have been addressed.” (Reviewer #3)

Second, all of the reviewers appeared to have read our manuscript and revision carefully and have made many constructive suggestions. We also appreciate Reviewer #1 for his/her detailed comments on behalf of Reviewer #2. The major concerns include: (1) Making fair comparison with other metrics (Reviewer #1); (2) Improving textual interpretation and acknowledging the value of other metrics (Reviewer #1 and #2); (3) Removing the overstatements and explaining the limitations of methylation concurrence (Reviewer #1 and #2). Therefore, we have performed additional analyses (**9 new figures**) and tuned the language, which together we believe strengthen our findings and clarify the role of our metric relative to other read-level metrics:

(1) Making fair comparison with other metrics

- We re-calculate methylation concurrence under the same reads-filtering setting as the calculations of methylation variation. The results confirm that methylation concurrence still performs better than average methylation and methylation variation (**New Supplementary Fig. 19**).

(2) Acknowledging the value of other metrics

- We clarify that, similar to methylation concurrence, methylation variation scores also show a negative correlation with expression.
- For the correlation with gene expression change (**Supplementary Fig. 8**), we note that the differences of average methylation and methylation variation show a negative correlation in half of their dynamic range.
- We agree that other metrics have their unique value in depicting methylation dynamics, and present methylation concurrence as another unique value, not a replacement.

(3) Removing overstatements and explaining the limitations of methylation concurrence

- We note that methylation concurrence is more sensitive to the changes in DNMT than TET activity, suggesting additional mechanisms may be involved for its negative regulation. Additionally, we present **New Supplementary Fig. 5c** as an example to explain this point.
- We remove the statement of ‘integrative analysis’ in Introduction.
- For the example gene *ZBTB16*, we change the wording of ‘strong association’ to ‘visible association’.

(4) Other improvements

- We change the strategy of selecting control genes for aCanyon target genes. (**New Fig. 4b, 4c, 4d, and 4e**)
- We add the summary plot in **New Fig. 4f** to show the Canyon distribution around the TSS.

In light of these substantial improvements, we are optimistic that all of the reviewers' concerns have been satisfactorily addressed. Please see below for a detailed **point-by-point response** (the reviewers' original comments are in blue).

Response to Reviewer #1:

Reviewer #1 (Remarks to the Author): I would like to thank the authors for all of their effort to add texture to this manuscript. I agree that "concurrency" is a more appropriate term for the metric in question and feel the additional figures, text, and analyses add improved resolution and depth that will benefit reception by the field. All told, I find the association between this new metric with gene expression, differential regulation across tumorigenesis, and the epigenetic regulation of unique local territories (canyons) to be intriguing, and there are certain benefits to this methodology compared to other read-level methylation-based conventions. However, I still feel the language of this manuscript overstates the conceptual advance of this method compared to others (often stating what it finds that other metrics do not and attributing it to mechanism), overinterprets the connection between read-level heterogeneity and the dynamic interplay between DNMTs and TETs specifically, and often assigns new functions from largely correlative observations. In most cases, my recommendation is that the method and findings are worth reporting, but that the language requires additional tuning. I believe this should heighten the impact of the manuscript by highlighting the value of this approach for finding and evaluating epigenetically dynamic regions, which I believe is its strongest selling point.

Response: We thank the reviewer for acknowledging the significance of the new method and our efforts in the last revision. Based on the reviewer's suggestions, we have tuned the language, removed the overstatements, and added new sentences in the Discussion section to clarify limitations. Please see the details under specific comments below.

I am still not convinced about the conceptual uniqueness of methylation concurrency compared to other read-level methylation metrics such as entropy or PDR (the proportion of discordant reads), which also prioritize local, read-level heterogeneity in methylation state as a read out for dynamic regulation. Instead, it seems more to me like concurrency has a more carefully bounded dynamic range, and some of this may differ only because the authors apply conventions to prior metrics out of expedience that they do not for their own.

Response: We agree with the reviewer that both methylation concurrency and methylation variation (PDR/Entropy/Epipolymorphism) scores contain information on local methylation heterogeneity. We also thank the reviewer for pointing out that methylation concurrency has a more carefully bounded dynamic range than the other metrics.

Epipolymorphism and methylation Entropy are based on the frequency of different methylation patterns (i.e., epi-alleles) on fixed windows of adjacent CpGs. A window with more CpGs will detect more diverse epi-alleles, thereby capturing more accurate heterogeneity information. For example, there are only 4 probabilities for a 2-CpG window, which is not enough to depict the different heterogeneity levels. Accordingly, a window of 3-CpGs will be better, as there are 8 probabilities. However, requiring more CpGs in a read will decrease the read-utilization. Therefore, these studies suggest using windows of 4-CpGs. This is likely in order to balance the trade-off between higher epi-allele diversity and higher read-utilization.

Methylation concurrence is not window-based and therefore does not have such limitations. Thus, it can utilize all reads. But the superior performance of methylation concurrence is not just because of its higher read-utilization. Please see our new analysis in **New Supplementary Fig. 19** which will be explained under the following comments.

For example, in Supplementary Figure 2, it appears to me that the positive relationship between entropy/epipolymorphisms and expression holds for the majority of promoter regions, but these are “squished” to the far left of the plot (the LOWESS regression also suggests a positive correlation within this portion of the plots, where the majority of the data is located). These trends (and the correlations) appear born out by the minority of outlier promoters that dominate the dynamic range of these estimates. To me, concurrence does not look so different, other than that these outliers are more constricted. The relationship between concurrence and entropy to gene expression is also highly similar in Figure 2b and Supplementary Figure 7, and the overall enrichment change in the boundaries of canyons is also similar in Supplementary Figure 15.

Response: As we mentioned above, methylation variation scores (e.g., PDR, Entropy, Epipolymorphism) only take reads covering at least 4 CpGs, but methylation concurrence and average methylation take all reads. To make a fair comparison, we re-calculate methylation concurrence and average methylation under the same reads-filtering setting as the calculations of methylation variation, in which only reads covering ≥ 4 CpGs are used.

As indicated in **New Supplementary Fig. 19a**, across the whole genome, reads covering ≥ 4 CpGs only account for 20.5% of total reads. Further, this proportion is clearly associated with CpG density of a particular genomic region. It is 83.6% at high-CpG promoters (HCP), 66.1% at intermediate-CpG promoters (ICP), and 16.5% at low-CpG promoters (LCP). As shown in **New Supplementary Fig. 19b**, the calculations of methylation concurrence only using reads covering ≥ 4 CpGs are similar to the calculations using all reads. The Pearson’s correlation between them is 0.94 for promoters, 0.92 for gene-body regions, and 0.86 for enhancers. The average methylation is also stable in this case.

The original Supplementary Figure 2 shows the correlation between DNMT/TET binding and methylation metrics. Here we re-analyzed this using methylation concurrence and average methylation based on the filtered reads. As indicated in **New Supplementary Fig. 19c**, using the same reads as methylation variation (shown in **Fig. 1d**), methylation concurrence is still highly correlated with the DNMT-TET joint regulation score, and performs better than the other metrics. We agree that methylation variation scores also show such a trend at the far-left part. But, as the reviewer mentioned before, methylation concurrence has a more carefully bounded dynamic range, which means it shows the global pattern in a more unbiased way.

We also re-analyzed the correlation between gene expression and methylation concurrence based on the filtered reads. As indicated in **New Supplementary Fig. 19d**, using the same reads as methylation variation, methylation concurrence is still consistently negatively correlated with gene expression, and outperforms average methylation (shown in right column of **New Supplementary Fig. 19d**) and methylation variation (shown in right column of **Fig. 2b**). We agree that, although less significant, methylation variation is also negatively correlated with gene expression. To clarify this point, we add the following sentences in the Discussion section:

“Another advantage of methylation concurrence is that it can utilize all reads, while the

methylation variation scores are window-based and only take reads covering at least 4 CpGs. To make a fairer comparison, we re-calculate methylation concurrence using the same reads used by methylation variation. The results confirm that methylation concurrence still performs better than average methylation (Supplementary Fig 19) and methylation variation (Fig 1d). It is worth noting that although its global correlation with expression is superior to the other compared metrics, methylation concurrence is not a perfect indicator for all genes. Other metrics have unique value in depicting methylation dynamics (Supplementary Fig 20c) and sometimes show a similar trend with methylation concurrence (Fig 2b and Supplementary Fig 15). In the future, a better indicator may be proposed by integrating different methylation metrics.”

New Supplementary Fig. 19. The calculations of methylation concurrence and average methylation only using reads covering at least 4 CpGs.

(a) The percentage of reads covering at least 4 CpGs in whole genome and different genomic features in CD3+ T-cells. HCP, high-CpG promoters; ICP, intermediate-CpG promoters; LCP, low-CpG promoters. (b) In CD3+ T-cells, the calculations of methylation concurrence (left) and average methylation (right) only using reads covering ≥ 4 CpGs (X-axis) are highly correlated with the calculations using all reads (Y-axis). Pearson's correlation was calculated. P-values were calculated by the two-tailed correlation test for Pearson's correlation. The diagonal line (slashed) indicates Y equal to X. (c) In mouse ESCs, the promoter methylation concurrence calculated using reads covering ≥ 4 CpGs significantly better correlates with 'DNMT3A1-TET1 joint regulation score' than average methylation does. Spearman's rank correlation was calculated. P-values were calculated by the two-tailed correlation test for Spearman's correlation. (d) The promoter/gene-body/enhancer methylation concurrence calculated using reads covering ≥ 4 CpGs are strongly negatively correlated with gene expression level in CD3+ T-cells, and this correlation is stronger than that of the average methylation. Spearman's rank correlation and Pearson's correlation were calculated based on all data points. P-values were calculated by the two-tailed correlation test. LOWESS lines were plotted to describe the relationships between variables (indicated by red curves).

If I am reading the methods correctly, the calculations for entropy, epipolymorphism and PDR all pre-filter on reads where at least 4 CpG strings can be measured in phase. As the authors show in their new track figures, that would suggest that their new metric benefits by considering and including reads in a different manner than for these other approaches. However, this is only a difference in convention and not in concept (this is also stated by the authors in their comments to Reviewer 2 regarding the OGT locus). Getting 4 CpGs in the same sequencing read is also far less likely outside of CpG islands, which would mean that more global claims are displayed as a fair comparison, but actually "mask" or remove substantial data for these other metrics. This is basically accommodating somewhat arbitrary conventions for how they are typically used and are not fair comparisons.

Response: We also agree with the reviewer that comparison using different read-filtering settings are inequitable. Therefore, we conduct the fair comparison in **New Supplementary Fig. 19c** and **19d**, in which we re-calculate methylation concurrence using the same pre-filtered reads as methylation variation. According to these results (showed above), the performance of methylation concurrence is still better under the same read-filtering as methylation variation, which means that the advantages of methylation concurrence are not due to different read-filtering conventions.

Entropy, PDR, and epipolymorphisms are conceptually quite similar to concurrence. At the very least, I think the language of the text should more carefully establish the fact that, as a metric, concurrence is designed to maximize detection of heterogeneous methylation according to a similar concept as these other approaches, as opposed to suggesting that these other methods don't cover very similar intellectual territory. This is particularly important because the language currently suggests that the authors treat each metric fairly, as opposed to applying empirical conventions to historical metrics that compromise the comparison to their new one.

Response: We agree with the reviewer that methylation concurrence maximize the data utilization to detect the local methylation dynamics. Through the fair comparison (**New**

Supplementary Fig. 19c and 19d), we confirmed the advantages of methylation concurrence under the same read-filtering setting. But we also agree that other metrics show similar trend in some respects. Therefore, we add further explanation in the Discussion section. Please see the bottom of **Page 3** in this letter.

Additional Comments:

Within the abstract, the second sentence is ambiguous and overestimates the audiences understanding of the term “concurrency.” Possibly this could be changed to something like “Although the dynamics of either methylation or demethylation have been intensively studied in the past decade, the direct effects of their interaction on gene expression remains elusive.”

Response: Thank you for the suggestion. The second sentence in Abstract has been modified as reviewer suggested.

As to the major point above, Supplementary Figure 1b should clarify the meaning of N/A to indicate that Shannon entropy and epipolymorphisms are not calculated for this right-most class of reads (though they could be if a different convention were applied). Similarly, would PDR as calculated in this manuscript consider these 3 and 2 CpG strings (it does not appear to be the case from the methods)? These differences in convention may also impact the ability for metrics like Entropy to be fairly applied to LCP-like loci as shown in Supplementary Figure 9 (≥ 4 CpG containing reads will be more rarely captured in this regime).

Response: Thank you for the suggestion. Although the other metrics can be applied on different lengths of epi-alleles (e.g., 2-CpG or 3-CpG), the previous studies suggest using 4-CpG windows, which is a balance between higher epi-allele diversity and higher read-utilization. So, we follow their suggestion in the current study. In the legend of **Supplementary Fig. 1b**, we add explanation for the N/A in this figure. We would also like to apologize for the typo in the last row. The calculation for PDR in the right-most example should be N/A.

Supplementary Fig. 1b. The calculations of different methylation metrics using examples of bisulfite-seq reads, which cover a 4-CpG region. As suggested in the previous studies, methylation Entropy, Epipolymorphism, and PDR only take reads covering at least 4 CpGs. Therefore, in the right-most example, the calculation of methylation variation is not available (indicated by N/A).

The authors claim to have conducted an integrative analysis of 85 methylomes and 44 transcriptomes within the introduction, but the manuscript itself still favors single samples or

limited data sets to make select points (a major comment of the first review that is not really addressed). The authors may have processed this many samples for their new metric, but this is not the same as an integrative analysis and misleading about what the authors have actually done. The language of this should be changed and possibly included moved to the discussion as a provided resource to the community.

Response: Thank you for the suggestion. Although we have validated our findings with multiple samples, we agree that the description of “integrative analysis” is not accurate. Thus, we have removed this statement in Introduction.

I very much appreciate the change in language to highlight the intriguing nature of the authors’ findings without oversuggesting a mechanistic connection. However, across their functional comparisons, the change in concurrence is much more convincing for DNMT3 KO than TET KO contexts, suggesting additional factors specifically impact the negative regulation of DNA methylation. The new text in the conclusions around these points should highlight this particular discrepancy. For example, “Notably, the change in concurrence only requires that DNA methylation be heterogeneous on the same molecule, and is more sensitive to changes in DNMT than TET activity, suggesting additional mechanisms may act to negatively regulate this modification.”

Response: Thank you for the suggestion. We agree that methylation concurrence is more sensitive to the changes in DNMT than TET activity. Therefore, we change the statement as below. And we also include Reviewer Figure 5 in **Supplementary Fig. 5c** and use this as an example to explain that methylation concurrence is not fully depleted in *Tet* KO.

“In agreement with the mouse data, methylation concurrence appears more sensitive to changes in DNMT activity than TET activity, suggesting additional mechanisms may be involved in its negative regulation. For example, some unmethylated regions in WT become partially methylated in TKO, so the methylation concurrence is elevated rather than depleted (Supplementary Fig 5c).”

The association between changes in gene expression and DNA methylation as calculated by concurrence, average methylation, or entropy are not particularly convincing (Supplementary Figure 8, requested in review). The comparison is between lung cancer and tumor-adjacent normal tissue from TCGA and the statements justified by Pearson correlation across the full dynamic range. Even visual inspection suggests that the relationship between promoter DNA methylation and expression changes is not cumulatively predictive. As most promoters that change their methylation status appear to gain DNA methylation as part of transformation, there is a clearer enrichment for the average methylation changes in the bottom right quadrant (increasing methylation that corresponds to gene repression). This strikes me as the most convincing result of this test. However, the language of the text associated with this figure instead concludes this is supportive of concurrence as the most predictive DNA methylation metric for gene expression. I think that is a very cursory statement that overstates the conclusions drawn from this exploration.

Response: The **Supplementary Fig. 8** indicates that, across their full dynamic range, Δ concurrence is better correlated with gene expression change than Δ average and Δ entropy. Δ average and Δ entropy only show a negative correlation in half of their dynamic range ($\Delta > 0$). This suggests that methylation concurrence reveals the global pattern in a more unbiased way.

However, we agree that an increase in average methylation is still a good indicator of gene repression. Therefore, we have modified the statements as below.

“In addition to being associated with baseline gene expression level, the differences (Δ) of methylation concurrence are also negatively correlated with gene expression changes (Supplementary Fig 8a) across its full dynamic range. In contrast, the differences of average methylation (Supplementary Fig 8b) and methylation variation (Supplementary Fig 8c) only show a negative correlation in half of their dynamic range ($\Delta > 0$). Altogether, the methylation concurrence ratio is a more unbiased predictor of gene expression than both the average methylation and the methylation variation.”

The closing statement of the canyon section should be reworded. Although the authors “redefine” or “rediscover a- and pCanyons using the concurrence metric, I imagine they could be more easily and precisely identified using gene expression (or Polycomb enrichment). For example, “Together, concurrence shows that the dynamics of methylation and demethylation are unique for a- and pCanyons and strongly associate with additional distinguishing features of their regulation.”

Response: Thank you for the suggestions. We have modified the closing statement of Canyon section as below.

“Together, methylation concurrence shows that the dynamics of methylation and demethylation are unique for aCanyons and pCanyons and strongly associate with additional distinguishing features (e.g., chromatin accessibility) of their regulation.”

All told, I find the added control features to the Canyon section a little unconvincing. The heightened negative Spearman correlation between gene body methylation and expression in aCanyons in Figure 4 could very well be the effect of the difference in the dynamic range for gene expression in this set compared to the “random highly expressed” background (Figure 4b). Canyons are pre-selected as large undermethylated regions, which did not utilize concurrence in their discovery (only their characterization). This difference between the Spearman correlations in aCanyons versus very highly expressed genes could effectively reflect bias between the aCanyon expression distribution and that of the “very highly expressed gene” sets.

Response: Instead of the “random highly expressed genes”, we change the strategy of selecting the control gene set for aCanyon target genes. For each aCanyon target, we select the non-Canyon target gene which has the closest expression value to it. In this way, we include a new control (Ctrl) gene set which features a similar expression distribution as aCanyon target genes (**New Fig. 4b**). As indicated in **New Fig. 4d**, this Ctrl group presents the classical regulating model, in which gene expression negatively correlates with promoter methylation but positively correlates with gene-body methylation. In contrast to the classical model, aCanyon target genes present a unique regulating model.

New Fig. 4 (b) Expression of aCanyon target genes (red, n=1,212), pCanyon target genes (blue, n=438), control group which features a similar expression distribution as aCanyon Targets (purple, n=1,212), and control group of randomly selected lowly expressed genes (gold, n=1,000). **(d)** Spearman correlation between gene expression and average methylation of 100bp-bin in gene regions. Totally 120 bins from -2kb to 10kb were measured. **(e)** The profiles of average methylation for each gene group. The standard deviations are indicated by the width of the shaded area.

Because the authors flip from defining canyons to showing behavior centered around TSSs, I would like a third set of plots below Figure 4d and 4e that shows where the canyon boundaries are in relationship to the TSS and gene body (eg. %fraction Canyon on the y axis).

Response: Thank you for the suggestion. We add the summary plot in **New Fig. 4f** to show the Canyon distribution around TSS.

New Fig. 4 (f) The Canyon distribution around the TSS. In each position from TSS-10kb to TSS+10kb, the percentage of aCanyon (red) or pCanyon (blue) target genes covered by Canyon is shown on the Y-axis.

The authors should also consider the strength of their conjecture about the relationship versus correlation of a- and pCanyons with the affinity of different TFs for methylated DNA (which is derived from in vitro Selex experiments without additional chromatin contexts that likely dictate binding). The speculation of how the binding of these factors might impact changes in gene expression ignores the fact that many of the pCanyon enriched factors (Ezh2, Suz12, Rest) are associated with repressive complexes. Again, the fact that these canyons have differential enrichment of methyl-sensitive TFs is novel and can be reported, but the conclusions drawn from that association should mostly end with the idea that these classes appear to be differentially regulated even in the steady state.

Response: Thank you for the suggestion. Repressive factors such as EZH2, SUZ12, and REST are only included in the ChIP-seq analysis (**Supplementary Fig. 16a** and **16c**), but not included in the SELEX data analysis (**Supplementary Table 4**). Therefore, we don't know whether their 5mC preferences are associated with Canyon enrichment. However, we agree that the conclusion drawn from the association (**Fig. 4g**) is only suitable for transcription activators rather than repressors. Therefore, we clarify this limitation in Discussion as below:

“Although we have confirmed this observation by analyzing in vitro SELEX data of the 5mC preferences of TFs, it is worth noting that current model (Fig 4g) can only explain the association between methylation change and binding of transcription activators but not repressors. In the future, further efforts are required to elucidate this mechanism.”

Similarly, in Supplementary Figure 15, it's not clear to me how the random background was readjusted to make the claim that within-group interactions are higher in pCanyons than aCanyons, but the fact that the adjusting for length weakens the claim should be cause enough for caution. The authors should at least be clearer about how they select background features in their methods.

Response: To better explain how the control regions are generated, we add the following descriptions in the Methods section.

“To exclude Canyon length bias, we also checked the chromatin interactions in two control groups. Using the 'shuffleBed' function in bedtools, we generated a random genomic region for each Canyon whose length is equal to the Canyon. Then, the random regions whose lengths are equal to aCanyons were designated as the control group of aCanyons, while the random regions whose lengths are equal to pCanyons were designated as the control group of pCanyons.”

My comments that spurred Reviewer Figure 3 were related to the idea that there are going to be some contexts where concurrence will not be relevant for detecting or predicting transcriptional activity, but other metrics might be. For example, very highly methylated loci (promoters associated with many germline genes, certain retrotransposon classes such as IAPs that are highly DNA methylation dependent, or imprinted genes) will have a low concurrence precisely because they are very highly methylated and effectively maintained in a repressed state by DNMT1. The scatterplot of promoter DNA methylation and concurrence (top left-most plot in Reviewer Figure 3) highlights this point: a discrete class of promoters show low concurrence, are not expressed, and are very highly methylated. A very valuable next step for these authors or the field would be to work out ways to integrate different metrics that appear to capture distinct dynamics. It would be appropriate for this manuscript to both highlight this key point as well as note the potential for integrating different metrics as part of their Discussion.

Response: Thank you for the suggestions. Although methylation concurrence has a better correlation with gene expression globally, we agree that it is still not a perfect indicator of all genes. As the reviewer mentioned, there may be a better solution in future by integrating different metrics. Therefore, we include Reviewer Figure 3 in **Supplementary Fig. 20c** and add the following sentences in Discussion to highlight this point.

“It is worth noting that although its global correlation with expression is superior to the other compared metrics, methylation concurrence is not a perfect indicator for all genes. Other metrics have unique value in depicting methylation dynamics (Supplementary Fig 20c) and sometimes show a similar trend with methylation concurrence (Fig 2b and Supplementary Fig 15). In the future, a better indicator may be proposed by integrating different methylation metrics.”

Additional comments related to those made by Reviewer 2:

Reviewer 2 seems to share a similar skepticism that the differences between methylation concurrence and other read-level methylation metrics are not as dramatic as the authors claim, and in several instances, required additional “raw data” displays such as the new Supplementary Figure 3. These figures highlight how some of these discrepancies appear to be algorithmic and not conceptual, and are not always in alignment with author statements. For example, the reviewer highlights a peak in increased methylation concurrence within the Ogt locus that is not captured by variation. The authors address this concern by saying that the difference between variation and concurrence is in part explained by which reads are considered by the entropy algorithm, whereas concurrence considers many more reads. In this manner, many of the claims by the authors feel misleading for similar reasons as highlighted above. A reader would believe that the concept of entropy or read-level variation is not adept at capturing the same changes as concurrence, when in reality at least a part of the difference could simply be that the authors are including or excluding different data according to conventions (which are not applied equivalently to their own). Certainly these reads with fewer than 4 CpGs are exhibiting “variation” and show “increased entropy,” and these terms have meanings beyond the algorithmic convention applied by the authors.

Response: To make a fair comparison between methylation concurrence and variation scores, we add new analysis in **New Supplementary Fig. 19**, in which methylation concurrence is re-calculated using the same filtered reads (covering at least 4 CpGs) as methylation variation. As indicated in **New Supplementary Fig. 19c** and **19d**, methylation concurrence is still better correlated with the ‘DNMT-TET joint regulation score’ and gene expression. These results indicate that the advantages of methylation concurrence are not simply due to its higher read-utilization.

The authors should think more carefully about how to distinguish their metric and compare/contrast it to others more clearly. There is value in it, but there’s no need to discount these other methods in their entirety.

Response: With the new analysis in **New Supplementary Fig. 19**, we confirmed the advantages of methylation concurrence by a fair comparison with other metrics. But we agree that methylation concurrence is still not a perfect metric and other metrics still have their unique benefits. As we mentioned on the bottom of **Page 3** in this letter, we explained this point in Discussion.

Similarly, the Reviewer asks the authors to explain why the Tet KO cells do not show complete loss of concurrence. Notably, the authors use these KO data to experimentally justify their metric as representing conflicting methylation and demethylation within cells by DNMTs and TETs. Reviewer Figure 5 highlights that there are in fact very convincing instances where read-level heterogeneity increases in a manner that cannot be explained by

the purported biochemical model posited by the authors. At the very least, the authors should account for this, include these figure panels, and explain them.

Response: Thank you for the suggestion. We include the Reviewer Figure 5 into **Supplementary Fig. 5c** to explain that Tet KO does not show complete loss of concurrence, which also suggest that additional mechanisms may be involved for its negative regulation.

These and other instances highlight the need for more careful wording of the manuscript in general. In the track provided for the tumor suppressor ZBTB16, the authors note a strong association between high concurrence and transcriptional repression, though the values and nature of the trend are not particularly convincing (an increase from 0.124 to 0.232 for a metric whose dynamic range is only so clear, New Supplementary Figure 11a).

Response: Thank you for the suggestion. We changed the wording of 'strong association' to 'visible association'.

Response to Reviewer #3:

Reviewer #3 (Remarks to the Author): All comments have been addressed.

Response: We thank the reviewer for acknowledging our efforts in the last revision.

REVIEWERS' COMMENTS

Reviewer #1 (Remarks to the Author):

I would like to thank the authors for their diligent efforts to clarify and refine the language and analysis of their new methylation concurrence metric. All told, despite the occasional disagreement about interpretation, I find the manuscript to be much improved, fairer in the evaluation of this metric versus others and clearer about the extent of its interpretations. I have a few minor comments, the first two of which I consider particularly important to rectify:

The added language around the major differences between Concurrence and other metrics such as Entropy and Discordance in the discussion is very much appreciated, but I feel actually so relevant to its value that it should be highlighted as a motivational feature in the second paragraph of the introduction and the first section of the results. For example, the description of the small fraction of reads that inform window-based methods in Supplementary Figure 19a is highly compelling for concurrence as a metric that can be applied more universally across the genome to extract a heterogeneity-based estimate. The reader should be informed of these discrepancies/difficulties in applying current metrics as a practical advantage for concurrence early in the text. As part of this, I would like to see an expanded version of Figure 19a included as part of Supplementary Figure 1 alongside an evaluation of the percent of the genome that can be appropriately measured via concurrence for the same set of features (which still requires multiple CpGs to be captured in phase and will not necessarily happen for every CpG or read).

Similarly, there should be additional data or qualification within the manuscript about the inclusion and density of data as measured by concurrence versus variation, epipolymorphism or discordance (and this should be presented early). For example, I think the reader should be prepared to understand how many more measurements go into the calculation of concurrence compared to variation for essential features, particularly promoters, gene bodies, and canyons. This includes the number of elements that can be measured as well as the mean or median coverage (and variation) for those elements. For example, how many 4-mer tiles on average go into an entropy measurement for a promoter compared to concurrence? Also, in any situation where concurrence is being evaluated against another read-level based metric (such as Supplementary Figures 7 or 9), the number of included features should be stated to highlight how the concurrence measurements are intrinsically more data dense.

Because this manuscript describes an analytical method and itself contains no new experiments, the suggestions in my first two paragraphs seem like appropriate aspects to include. It doesn't strike me as unfair to ask that the authors include in their introduction why entropy and other metrics may be limited by resolution or convention (requires 4 CpGs to be captured on the same read) and how their method circumvents this. They should also include in their early extended data more detailed quantification of what concurrence recovers from sequencing data that these other metrics omit (eg. how many promoters are captured at a given read depth, how many independent measurements go into calculating a single promoter value, how do these compare to other metrics?). This would go a very long way in guiding the reader to evaluate all forthcoming claims throughout the manuscript.

The hypothetical model around the methyl-TF data in the last paragraph before the conclusion still seems a little difficult to resolve. pCanyons speak to a specific form of regulation by Polycomb, meaning pCanyon genes will, in at least one cell type, be highly expressed and pivot to the epigenetic architecture of transcription: low promoter methylation and gene body methylation. It could be that the higher preference for methylation within pCanyon TFs speaks to their role in transcriptional induction (binding a silent state de novo and recruiting activation machinery), whereas many (if not most) aCanyons are constitutively active housekeeping genes and enrich for core promoter factors such as TBP that bind unmethylated promoter sequences to support ongoing transcription. The current speculative model focuses on transcriptional dynamics within these canyons without presuming the dynamic nature of canyon's epigenetic state as transcription changes. Personally, I would rather have seen some analysis on this point (eg. "how does concurrence capture the change in pCanyon regulation when the embedded gene is transcribed" or "how constitutive is aCanyon versus pCanyon epigenetic status across cell types") but understand

that this makes sense as part of follow up investigations. At minimum, some of these ideas could be included towards the end of the section, or the model for how TF binding might relate to concurrence be further refined according to the distinct identity of a and pCanyon genes.

The new Figure 4f is sufficient, but I was asking that the actual canyon boundaries be highlighted underneath the TSS composites in Figures 4d and e using the same display strategy. Presumably, because DNA methylation is increasing, you'd expect the average boundary to resemble the average methylation track. Similarly, I expect there may also be very clear associations between gene length or CpG density for Canyon genes that are not considered when selecting the random backgrounds. I realize that analyzing these additional features would require substantially more work, but highlighting how they might impact or explain the disparity in concurrence between aCanyons and background over the gene body could be quickly stated.

We appreciate the reviewers as they clearly acknowledged our efforts in the last revision. The reviewers have read our manuscript and revision carefully and have made many constructive suggestions, which help to improve this study substantially. Please see below for a detailed **point-by-point response** for the remaining questions (the reviewers' original comments are in blue).

Response to Reviewer #1:

Reviewer #1 (Remarks to the Author): I would like to thank the authors for their diligent efforts to clarify and refine the language and analysis of their new methylation concurrence metric. All told, despite the occasional disagreement about interpretation, I find the manuscript to be much improved, fairer in the evaluation of this metric versus others and clearer about the extent of its interpretations. I have a few minor comments, the first two of which I consider particularly important to rectify:

Response: We thank the reviewer for acknowledging the significance of the new method and our efforts in the last revision.

The added language around the major differences between Concurrence and other metrics such as Entropy and Discordance in the discussion is very much appreciated, but I feel actually so relevant to its value that it should be highlighted as a motivational feature in the second paragraph of the introduction and the first section of the results. For example, the description of the small fraction of reads that inform window-based methods in Supplementary Figure 19a is highly compelling for concurrence as a metric that can be applied more universally across the genome to extract a heterogeneity-based estimate. The reader should be informed of these discrepancies/difficulties in applying current metrics as a practical advantage for concurrence early in the text. As part of this, I would like to see an expanded version of Figure 19a included as part of Supplementary Figure 1 alongside an evaluation of the percent of the genome that can be appropriately measured via concurrence for the same set of features (which still requires multiple CpGs to be captured in phase and will not necessarily happen for every CpG or read).

Response: We thank the reviewer's suggestion of highlighting the higher read-utilization of methylation concurrence and the limitations of methylation variation methods. Thus, we add the following sentence to the 2nd paragraph of the introduction.

"However, to capture more accurate heterogeneity information, these methylation variation quantifications require at least 4-CpGs covered by each sequencing read, and thus can only utilize ~20% of total reads in the genome (Supplementary Fig 1)."

To emphasize this point earlier, we reorganized some of the figures. As suggested by the reviewer, Supplementary Fig 19a is moved to **Supplementary Fig 1c**. The statistic of "Canyon" elements is also appended to the figure. To illustrate whether methylation concurrence can measure more elements than other metrics, we add a new analysis in **New Supplementary Fig 1d**.

New Supplementary Figure 1. (c) The percentage of reads covering at least 4 CpGs in the whole genome and different elements in CD3+ T-cells. HCP, high-CpG promoters; ICP, intermediate -CpG promoters; LCP, low-CpG promoters. **(d)** The percentage of different elements detected by methylation concurrence, average methylation, or methylation variation scores.

The following sentences are added to the first part of the Results section to explain these figures.

“The methylation variation scores are window-based and only take reads covering at least 4 CpGs (only ~20% of total reads), while methylation concurrence has no such limitation and utilizes all reads (Supplementary Fig 1c). Furthermore, methylation concurrence detects more regulatory elements (e.g., promoters) than methylation variation (Supplementary Fig 1d).”

Similarly, there should be additional data or qualification within the manuscript about the inclusion and density of data as measured by concurrence versus variation, epipolymorphism or discordance (and this should be presented early). For example, I think the reader should be prepared to understand how many more measurements go into the calculation of concurrence compared to variation for essential features, particularly promoters, gene bodies, and canyons. This includes the number of elements that can be measured as well as the mean or median coverage (and variation) for those elements. For example, how many 4-mer tiles on average go into an entropy measurement for a promoter compared to concurrence? Also, in any situation where concurrence is being evaluated against another read-level based metric (such as Supplementary Figures 7 or 9), the number of included features should be stated to highlight how the concurrence measurements are intrinsically more data dense.

Response: We thank the reviewer’s suggestion. To answer the question of “how many more measurements go into the calculation of concurrence compared to variation for essential features”, we add the new analysis in **New Supplementary Fig 1d**. Please check the figures above.

Because the methylation concurrence is not based on the 4-CpG window and many reads it utilized do not contain 4-CpGs, we believe it is not proper to compare its data usage with other metrics based on the number of “4-mer tiles”. For this purpose, the statistics in **Supplementary Fig 1c** are more suitable. As the methylation concurrence utilizes all reads (100%), this plot clearly shows how many reads are omitted by methylation variation metrics.

Although methylation concurrence utilizes more reads (**Supplementary Fig 1c**) and detects more elements (**Supplementary Fig 1d**) than methylation variation metrics, in **Supplementary Fig 7 and 9**, we only use the elements which can be detected by all three methods (e.g., concurrence, average, and variation) to make fair comparisons. The numbers of included elements are indicated in related figure legends.

Because this manuscript describes an analytical method and itself contains no new experiments, the suggestions in my first two paragraphs seem like appropriate aspects to include. It doesn't strike me as unfair to ask that the authors include in their introduction why entropy and other metrics may be limited by resolution or convention (requires 4 CpGs to be captured on the same read) and how their method circumvents this. They should also include in their early extended data more detailed quantification of what concurrence recovers from sequencing data that these other metrics omit (eg. how many promoters are captured at a given read depth, how many independent measurements go into calculating a single promoter value, how do these compare to other metrics?). This would go a very long way in guiding the reader to evaluate all forthcoming claims throughout the manuscript.

Response: We agree with the reviewer's suggestions. As we mentioned before, we have added text in the Introduction and the Results sections to explain the limitations of published metrics. **Supplementary Fig 1c** and **1d** illustrate the percentage of reads and elements omitted by methylation variation metrics. To make fair comparisons, we only use the elements which can be detected by all three metrics (e.g., concurrence, average, and variation) for the expression correlation analysis. Thus, under the same sequencing depth, the included element numbers are the same for different metrics (**Supplementary Fig 9a**).

New Supplementary Figure 9a. Spearman correlations between methylation quantitative measures and gene expression at varying sequencing depths. Y-axis indicates the Spearman correlation between gene expression and three methylation quantitative measures of gene promoters in CD3+ T-cells. The X-axis shows the average sequencing depth of down-sampled bisulfite-seq data. The numbers of included promoters are indicated below.

The hypothetical model around the methyl-TF data in the last paragraph before the conclusion still seems a little difficult to resolve. pCanyons speak to a specific form of regulation by Polycomb, meaning pCanyon genes will, in at least one cell type, be highly

expressed and pivot to the epigenetic architecture of transcription: low promoter methylation and gene body methylation. It could be that the higher preference for methylation within pCanyon TFs speaks to their role in transcriptional induction (binding a silent state de novo and recruiting activation machinery), whereas many (if not most) aCanyons are constitutively active housekeeping genes and enrich for core promoter factors such as TBP that bind unmethylated promoter sequences to support ongoing transcription. The current speculative model focuses on transcriptional dynamics within these canyons without presuming the dynamic nature of canyon's epigenetic state as transcription changes. Personally, I would rather have seen some analysis on this point (eg. "how does concurrence capture the change in pCanyon regulation when the embedded gene is transcribed" or "how constitutive is aCanyon versus pCanyon epigenetic status across cell types") but understand that this makes sense as part of follow up investigations. At minimum, some of these ideas could be included towards the end of the section, or the model for how TF binding might relate to concurrence be further refined according to the distinct identity of a and pCanyon genes.

Response: Although our analysis has shown the distinct features of aCanyons and pCanyons, we agree with the reviewer that the current hypothetical model is not sufficient to explain their distinct dynamic regulations. More efforts are needed to investigate how methylation concurrence affects Canyon gene regulation in a biological context, e.g., tumorigenesis. As this is more appropriate to be part of follow-up studies, we add the following text in the last part of the Results section to emphasize this point.

"This model helps explain the distinct regulations of aCanyon and pCanyon targets in one particular cell type. However, how methylation concurrence captures their transcriptional changes from one condition to another is unclear and will need to be elucidated in follow-up investigations."

The new Figure 4f is sufficient, but I was asking that the actual canyon boundaries be highlighted underneath the TSS composites in Figures 4d and e using the same display strategy. Presumably, because DNA methylation is increasing, you'd expect the average boundary to resemble the average methylation track. Similarly, I expect there may also be very clear associations between gene length or CpG density for Canyon genes that are not considered when selecting the random backgrounds. I realize that analyzing these additional features would require substantially more work, but highlighting how they might impact or explain the disparity in concurrence between aCanyons and background over the gene body could be quickly stated.

Response: Because most Canyons spread from upstream of gene TSS to gene-body regions, if we use the same display strategy as in **Fig 4d** and **4e** (from -2kb to +10kb), many Canyons will be truncated. To show the complete distributions, we expanded the displayed regions from -10kb to +10kb of TSS in **Fig 4f**.

As the expression levels of aCanyon and pCanyon genes are dramatically distinct, it is the main confounding factor we must overcome when selecting the Ctrl genes. This is also suggested by the reviewer in previous revisions. However, we agree that other features, such as gene length and CpG density, are also important. In the future, substantial efforts are needed to systematically investigate these factors and how they are associated with Canyon genes' regulating, which we believe is beyond the scope of the current study.